# Young SINEs in pig genomes impact gene regulation, genetic diversity, and complex traits

Pengju Zhao[1,2], Lihong Gu[3], Yahui Gao[4], Zhangyuan Pan[5], Lei Liu[6], Xingzheng Li[6], Huaijun Zhou[5], Dongyou Yu[1,2], Xinyan Han[1,2], Lichun Qian[1,2], George E. Liu [4✉], Lingzhao Fang [7✉] & Zhengguang Wang [1,2✉]

Transposable elements (TEs) are a major source of genetic polymorphisms and play a role in chromatin architecture, gene regulatory networks, and genomic evolution. However, their functional role in pigs and contributions to complex traits are largely unknown. We created a catalog of TEs ($n = 3,087,929$) in pigs and found that young SINEs were predominantly silenced by histone modifications, DNA methylation, and decreased accessibility. However, some transcripts from active young SINEs showed high tissue-specificity, as confirmed by analyzing 3570 RNA-seq samples. We also detected 211,067 dimorphic SINEs in 374 individuals, including 340 population-specific ones associated with local adaptation. Mapping these dimorphic SINEs to genome-wide associations of 97 complex traits in pigs, we found 54 candidate genes (e.g., *ANK2* and *VRTN*) that might be mediated by TEs. Our findings highlight the important roles of young SINEs and provide a supplement for genotype-to-phenotype associations and modern breeding in pigs.

[1] Hainan Institute, Zhejiang University, Yongyou Industry Park, Yazhou Bay Sci-Tech City, Sanya 572000, China. [2] College of Animal Sciences, Zhejiang University, Hangzhou, Zhejiang 310058, China. [3] Institute of Animal Science & Veterinary Medicine, Hainan Academy of Agricultural Sciences, No. 14 Xingdan Road, Haikou 571100, China. [4] Animal Genomics and Improvement Laboratory, Beltsville Agricultural Research Center, Agricultural Research Service, USDA, Beltsville, MD 20705, USA. [5] Department of Animal Science, University of California, Davis, CA 95616, USA. [6] Shenzhen Branch, Guangdong Laboratory of Lingnan Modern Agriculture, Genome Analysis Laboratory of the Ministry of Agriculture and Rural Affairs, Agricultural Genomics Institute at Shenzhen, Chinese Academy of Agricultural Sciences, Shenzhen 518124, China. [7] Center for Quantitative Genetics and Genomics, Aarhus University, Aarhus 8000, Denmark. ✉email: george.liu@usda.gov; lingzhao.fang@qgg.au.dk; wzhguang68@zju.edu.cn

TEs and common repeats are ubiquitous sequences that can copy and insert themselves throughout the eukaryotic and prokaryotic genomes[1–7]. The movement of TEs is often accompanied by an increase in their abundance, comprising a large fraction of genomic sequences[8,9]. According to the mechanism of transposition, TEs can be generally classified into (1) RNA-mediated class I elements (retrotransposons), including long terminal repeats (LTRs), long interspersed nuclear elements (LINEs), and short interspersed nuclear elements (SINEs); and (2) RNA-independent class II elements (DNA transposons)[10]. TE classes could be further divided into distinct families or sub-families based on their age (active period) and DNA sequence characteristics.

At the predominant view of the 1960s–1990s, TEs were described as selfish or junk DNA[11]. Thanks to the availability of whole-genome sequence of various species and the ongoing development of bioinformatics tools[12–15], our knowledge of TEs has progressed at a fast pace. TEs are known to play an essential role in shaping genomic sequences and contributing to the diversity in genome size and chromosome structure[1,16,17]. Most TEs, in fact, are fixed, inactive, and not randomly distributed in the genome[18,19]. However, several TE families are still actively transposing and serving as a major source of genetic polymorphisms between individuals, such as the Alu, L1, and SVA TE families in the human genome[20].

It is evident in many species (e.g., humans and rice) that the impacts of active TEs on genome evolution are wide-ranging, including admixture, adaptation, footprints of selection, and population structure[21–24]. For example, the polymorphic TEs detected in the 1000 Genomes Project, consisting of 16,192 loci in 2504 individuals across 26 human populations, successfully recapitulated human evolution and captured the signal for positive selection on recent human TE insertions[20,25,26].

In addition to their direct influence on DNA sequence, there is also emerging evidence that TEs have important functional contributions to gene regulatory networks and epigenome variation. For instance, TEs can directly affect gene transcriptional structure by provoking various forms of alternative splicing, including exonization, exon skipping, and intron retention (3′ and 5′), to generate novel protein-coding sequences or premature ends[27–30]. TEs can disrupt the existing cis-regulatory elements, such as promoters, enhancers, and insulators, or provide novel ones[31–34]. They can also serve as a rich source of non-coding RNAs, including lncRNAs, circRNA, small RNAs, and microRNA targets[35–38]. Moreover, the silencing of TEs has a close connection with epigenetic regulatory mechanisms, such as DNA methylation, piRNA, histone modifications, and RNA interference[18,39–42]. Importantly, it has been reported that the complex interactions between TEs and epigenetic elements could allow for rapid phenotypic adaptation to environmental changes[40,42,43].

Pig (Sus scrofa), one of the earliest domesticated animals, is estimated to have been domesticated approximately 10,000 years ago in Asia and Europe independently[44]. It serves as an indispensable source of animal protein and an important biomedical model for humans[45,46]. Currently, a total of 22 pig assemblies are publicly available on NCBI[47–51], accompanied by the availability of massive high-throughput whole-genome sequences. These provide researchers with ideal materials to advance the current development of genomic research in pigs. However, the study of TEs in the pig genome is still in its infancy. A few previous studies have mainly focused on their diversity and distribution[47–51], yet the functional and evolutionary importance of TEs in pigs has largely been overlooked. In our recent study[52], we identified novel introgressions in Eurasian boars from Asian and European pig populations using SINE (PRE-1 subfamily) polymorphisms,

suggesting that a portion of TEs are still active in the current pig genome. However, these studies are far from sufficient to comprehensively understand the important roles of TEs in pigs.

In this study, we created a comprehensive and high-quality atlas of TEs so far in pigs and classified SINE families into four categories based on their ages using the newly built pipelines. We then systematically explored the genomic bias of these SINE categories by combining large-scale multi-omics data from 21 tissues, including three-dimensional chromatin architecture, chromatin accessibility, histone modifications, transcription factor binding sites (TFBS), and DNA methylation. We estimated the contribution of active SINEs to tissue-specific gene expression by cross-examining 3570 published RNA-seq samples from 52 tissues and 27 cell types. Furthermore, we created an atlas of SINEs using 374 whole-genome sequence data to study the roles of young SINEs in pig population admixture and local adaptation. The TE-mediated adaptation has been found in functional regions, such as the almost fixed dimorphic SINEs observed in laboratory-inbred Bama Xiang pigs at the upstream region of the *LEP* gene. Finally, by mapping these dimorphic SINEs to 4072 loci associated with 97 complex traits in pigs, we propose 54 candidate genes that might regulate complex traits through TEs.

## Results

**Composition of young SINE families in the pig genome**. To thoroughly detect TEs, we developed the Pig TE Detection and Classification (PigTEDC) pipeline (Fig. 1a) and applied it to the pig genome (*Sus scrofa* 11.1). This pipeline used a combination of similarity-, structure-, and de novo-based methods. We also classified all potential pig TEs into classes/superfamilies and families and derived their consensus sequences based on existing TE repositories (RepBase update and Dfam 2.0 databases).

Excluding nested TEs, we found 3,087,929 TEs occupying 37.9% (947 MB) of the pig genome. Two-thirds of TE copies (insert in the genome) were assigned to a specific family, with retrotransposons being the most common type (~90%). Similar to previous studies[47–51], LTR (9.25%), LINE (27.57%), and SINE (54.95%) were the most common retrotransposons, whereas DNA transposons only accounted for 8.12% of TEs (Fig. 1b). SINE being the most abundant in count but covering less genome size than LINE (Fig. 1c).

Out of 532 TE families, 65 (with over 3000 TE copies each) make up 84.6% of all classified TEs (Fig. 1d). These include PRE1f in SINE/tRNA (170,511 copies), MIR in SINE/MIR (45,927 copies), L1B-SSc in LINE/L1 (35,819 copies), and MLT1D in LTR/ERVL-MaLR (7866 copies). The stacking plots in Fig. 1e show the divergence distribution for superfamilies or families. Our analysis of pig TEs revealed two bursts at 10% and 30%, estimated to have occurred 20 and 60 million years ago, which is similar to the divergence distribution of TEs in the human genome[53]. Obviously, most TE families amplified around 70–50 Mya (divergence at 30 ± 5%). This was during the Paleocene Epoch (65–54 Mya) which created new ecological niches for surviving mammals, birds, reptiles, and marine animals[54]. The most recent burst of TEs was mainly related to the SINE/tRNA, LINE/L1, and LTR/ERV1 families. Among these, SINE/tRNA remains the most active in the modern pig genome[55–57]. Further exploring the ages of highly homologous subfamilies in SINE classes (Fig. 1f with an average divergence of 4%, labeled in purple), we found that 3 out of 26 SINE families (PRE1-SS, PRE0-SS, and PRE1a) were recently most prolific and shown to be polymorphic within pig breeds in a previous study[52], and thus considered as young SINE families.

We next analyzed young SINE families (PRE1-SS, PRE0-SS, and PRE1a) to classify them into subfamilies with high resolution.

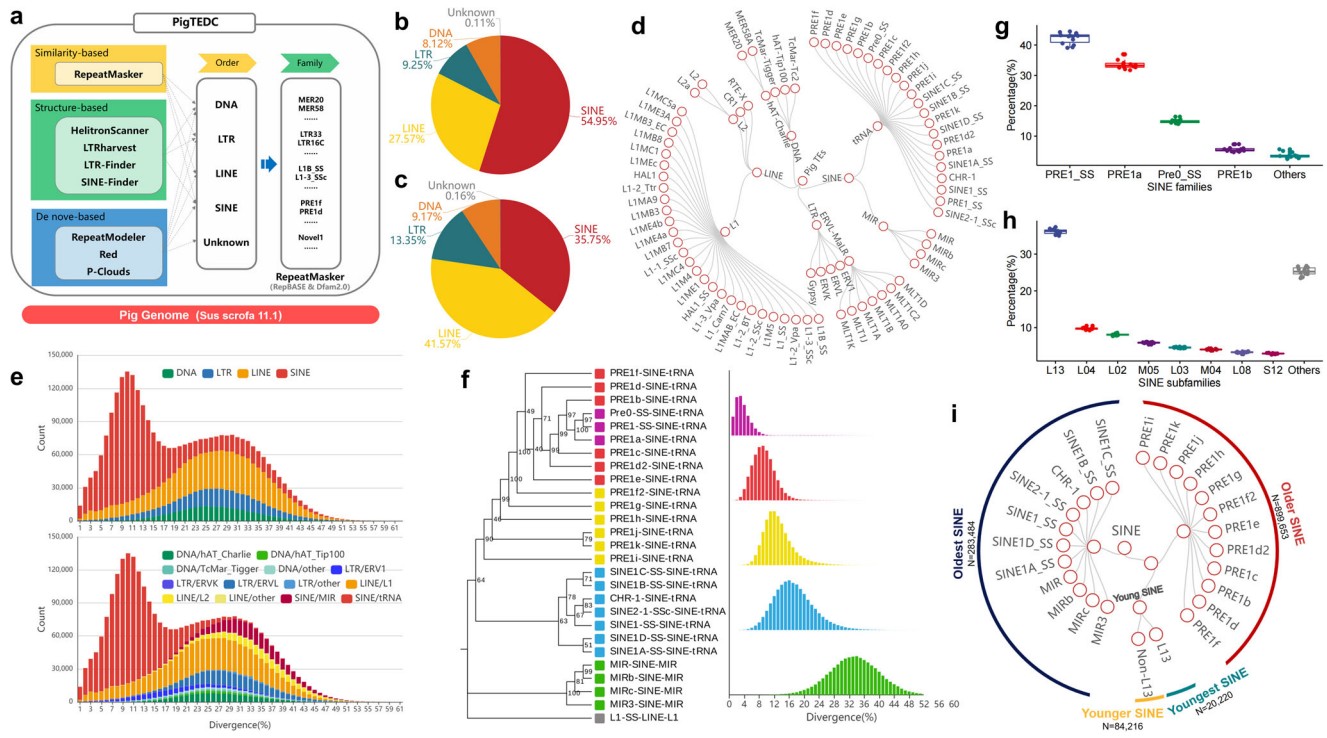

**Fig. 1 TE annotation and SINE classification in the pig genome. a** A schematic of the Pig Transposable Element Detection and Classification (PigTEDC) pipeline. It is composed of three TE detection approaches, which use similarity-, structure-, and de novo-based algorithms. **b** The proportion of TEs from different superfamilies in the count. **c** The proportion of TEs from different superfamilies in length. **d** Classification of pig TE superfamilies and families (≥3000 copies in each family). **e** Sequence divergence distribution for TE superfamilies (upper panel) and families (bottom panel) in the pig genome. Sequence divergence distributions are plotted in bins of 0.01 increments. **f** Phylogenetic tree and sequence divergence distribution for SINE families in the pig genome. On the right panel, the x-axis represents the divergence, and the y-axis represents the counts of the SINE families. **g** Boxplots display the proportion of genomic SVs formed by different SINE families. **h** Boxplots display the proportion of genomic SVs formed by different SINE subfamilies. See Supplementary Data 1 for their definitions. **i** Classification of pig SINE families based on their ages. The line inside each boxplot represents the median.

We examined all the full-length young SINEs from 14 publicly available pig genomes, identified 978,506 non-redundant young SINEs, and created their consensus sequence through multiple sequence alignment (Supplementary Figs. 1, 2). Subsequently, using a minimum spanning tree analysis, we recategorized the 90 subfamilies of SINE into 17 large (size >10,000), 12 medium (size ≥3000 and ≤10,000), and 61 small (size <3000) subfamilies (Supplementary Figs. 3, 4, Supplementary Data 1).

When comparing the locations of young SINEs with medium-length (range from 200 to 300 bp) structure variations (SVs) in 14 assemblies to the pig reference, we found that most polymorphic SINEs belonged to the PRE1-SS, PRE1a, and PRE0-SS families, accounting for an average of 90.75% of the medium-length SVs (Fig. 1g). Especially for the L13 subfamily accounted for an average of 36.15% of the medium-length SVs (Fig. 1h). Our findings indicate that only a certain group of recently active SINE subfamilies played a major role in causing SVs (about 250 bp long) in various pig breeds during recent evolution. To simplify our analysis, we classified all SINEs into four categories: youngest (L13 subfamilies), younger (SINE families other than L13 subfamilies and non-L13), older (non-young PRE families), and oldest (non-PRE families; ancient families) (Fig. 1i, Supplementary Fig. 5).

**Widespread roles of young SINEs in gene regulatory networks**. Previous studies have proposed that TEs may be co-opted into regulatory sequences of genes through diverse epigenetic mechanisms[58–61]. To test this, we examined how SINE subfamilies affect genome features such as 3D chromatin architecture[62], chromatin accessibility, histone modifications,

TFBS, and DNA methylation after only mapping unique reads (Fig. 2a).

We observed a highly enrichment of all SINEs in the A compartments (active), while there was a depletion in the B compartments (inactive) (Wilcoxon test, $P$-values $< 10^{-16}$, Fig. 2b, Supplementary Fig. 6). After dividing A/B compartments into topologically associating domains, it was observed that CTCF binding sites were enriched in the boundary regions of these domains. SINEs also exhibited a similar but slightly weaker trend of enrichment, while young SINEs showed higher but more variable enrichment compared to old SINEs (Fig. 2b, Supplementary Fig. 7). We next explored the distribution of SINE families on the chromatin accessibility and nucleosome positioning near transcripts using the published chip-seq (14 tissues) and MNase-seq (five tissues) datasets, respectively[63,64]. TE enrichment in chromatin showed age-specific patterns, with youngest and younger SINE families depleted from open chromatin but enriched near the nucleosome (Fig. 2c). Older SINE families were relatively highly enriched for open chromatin, especially in the stomach, adipose, and cerebellum. We further studied the relationship of SINE families with four active epigenetic marks (H3K4me1—primed enhancers, H3K4me3—enriched in transcriptionally active promoters, H3K27ac—which distinguishes active enhancers from poised enhancers, and H3K36me3—actively transcribed gene bodies) and two repressive marks (H3K9me3—constitutively repressed genes and H3K27me3—facultatively repressed genes). In Fig. 2c, we observed that most SINE families were underrepresented in all four active marks, consistent across tissues. However, young SINEs were highly enriched for H3K9me3, which indicates permanent repression,

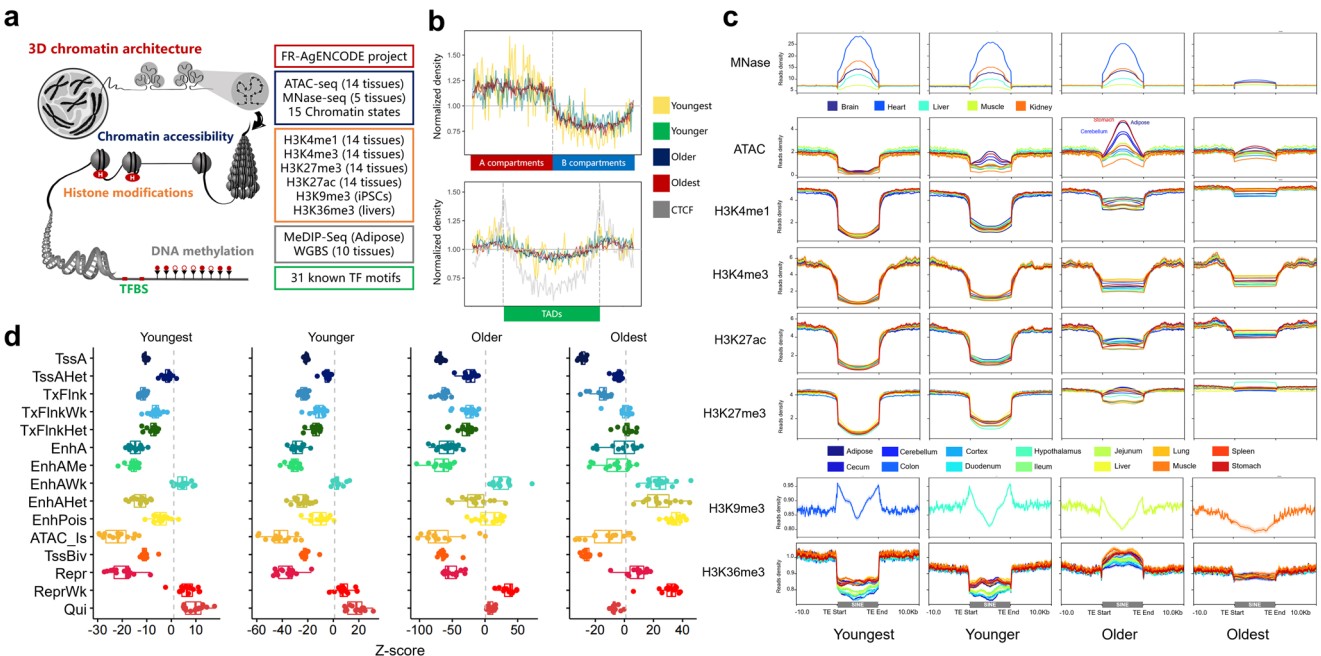

**Fig. 2 Distribution of SINE on pig genome and functional regions. a** The five types of genomic features used in this study included 3D chromatin architecture, chromatin accessibility, histone modifications, DNA methylation, and TFBS. **b** The distribution of SINE families between 3D chromatin architectures (Compartments A vs. B) and near topologically associating domains is examined. **c** The read density distributions of chromatin accessibility and histone modifications near transcripts were analyzed across four different SINE groups. **d** Boxplots display the enrichment of four SINE groups in 15 distinct chromatin states across 14 tissues. The line inside each boxplot represents the median.

but not for H3K27me3, which has development-dependent repressed characteristics[65]. In general, compared to old SINE groups, young SINE groups had a higher over-/under-representation for all histone modification types, especially the youngest group.

In addition, we investigated the enrichment of SINE families in 15 chromatin states across 14 tissues[66]. We observed that young SINE groups had lower enrichment in most chromatin states compared to old SINE groups. The degree of SINE enrichment in chromatin states was similar across all 14 tissues (Fig. 2d, Supplementary Data 2). The enrichment characteristics of different SINE groups can be divided into four distinct patterns based on their degree of enrichment (Fig. 3a). The oldest SINE group was highly enriched in most cases (80%, 168 out of 210 combinations of 14 tissues and 15 states), while the young SINE group showed three enrichment patterns in the remaining combinations (enlarged inset). In two of these patterns, only the youngest SINE group was highly enriched in TssAHet (flanking active TSS without ATAC) and EnhAWk (weakly active enhancer). Overall, SINE groups were depleted from active promoters and enhancers, except for weak TSS and enhancers. Young SINE groups were more depleted than old ones. This suggests that new SINEs may be silenced by histone modifications and DNA methylation, while older ones may be tolerated by the pig genome.

TEs carrying TFBS may contribute to the regulation of genes[67,68]. We performed motif enrichment analysis of SINEs to explore their possible contributions (Fig. 3b). In total, 31 TFBS were predicted to have binding motifs in at least one SINE family, mostly in old ones (96.8%). 83.9% of the TFBS related to open chromatin, indicating that young SINEs were rarely exapted into regulatory regions[69] and were repressed by less chromatin accessibility. The youngest SINE-specific TFBS related to the *ZNF148* gene has been shown to promote the development of a muscle phenotype[70]. Besides, three members of the RFX transcription factor family were amplified in both young and

older SINEs and involved in immune, reproductive, and nervous pathways[71]. For instance, *RFX1* and *RFX3* were found to be major histocompatibility complex (MHC) class II promoter binding proteins that functioned as trans-activators of the hepatitis B virus enhancer[72,73].

Given that TEs play major roles in gene expression regulation by shaping the epigenetic modifications[74], we analyzed the epigenetic states of SINE families by examining DNA methylation (MeDIP), density of CG (CpG) sequence contexts, and AT:GC content (CpG islands) (Fig. 3c). The results showed that almost all SINE families exhibited a depletion in genomic regions with CpG islands, and the young SINE families were more highly methylated than the old ones. Similarly, we found that CG methylation levels in SINE bodies, particularly in young SINEs, were higher than in their flanking regions across 10 different tissues. This corresponds to the enrichment of SINE families in H3K9me3 (Fig. 3d).

A previous study revealed that piRNAs play a major role in TE silencing via the ping-pong cycle in pig germline[75]. We further distinguished small non-coding RNAs into three classes to investigate the relationship between piRNA density and SINE families (Supplementary Fig. 8). Unlike siRNA and miRNA, piRNA was highly enriched for SINE-related sequences or sequence flanks, and there was a negative correlation between piRNA density and the age of the SINE subfamily. Our result agrees with findings in humans that young SINE families are more prone to piRNA targeting[76], which may be the main reason for their high methylation levels, as we observed above.

**Young SINE-associated transcriptome profiling in pigs**. TEs can modify host gene transcription directly by remodeling alternative splice events or providing cis-regulatory sequences[77–79]. To test this, we analyzed PacBio long-read iso-form sequences from 38 pig tissues to detect transcripts of SINE-associated exonization and alternative splice sites[80,81]. We

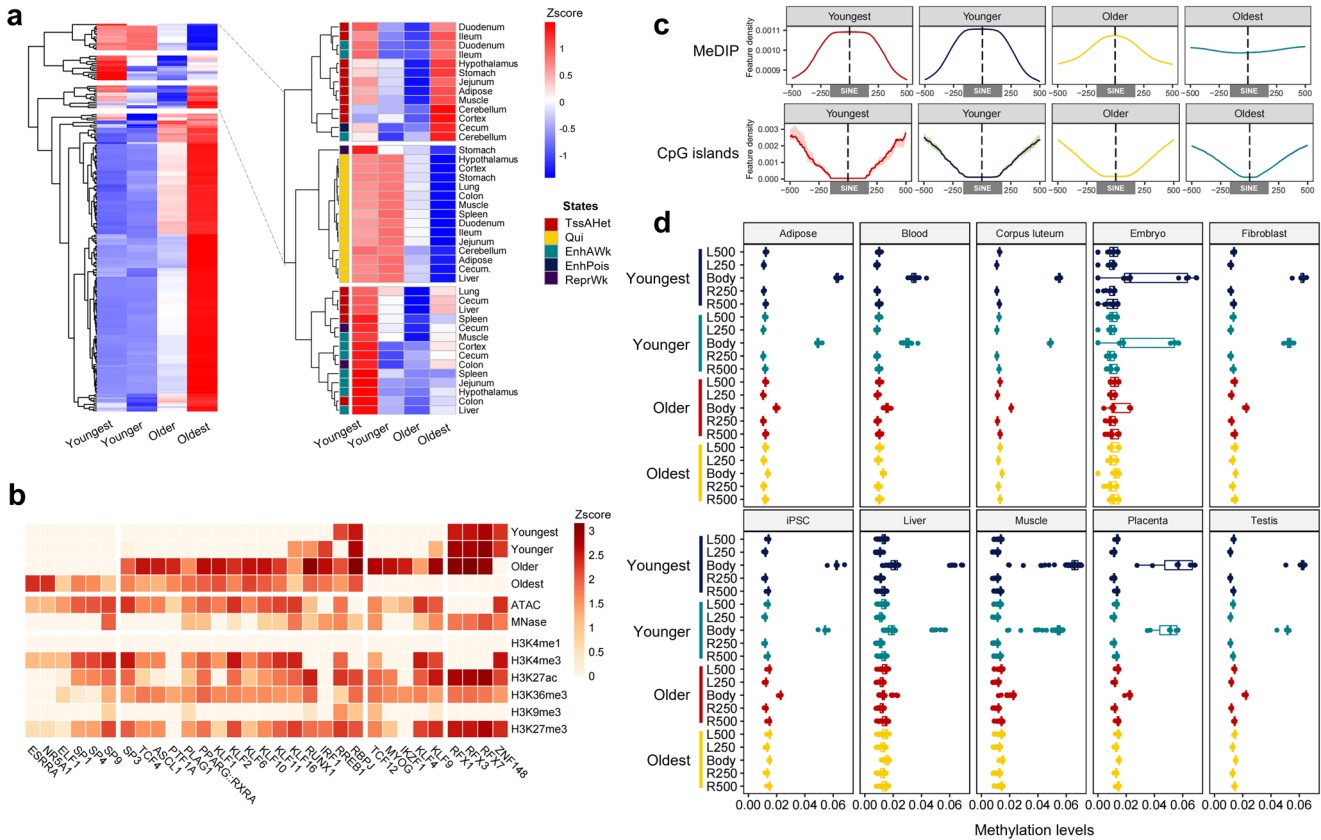

**Fig. 3 Enrichment of SINE in functional elements and methylation modification. a** Hierarchical clustering of enrichment patterns in 15 chromatin states for four SINE groups across 14 tissues (left panel). The heatmap shows three distinct enrichment patterns for high enrichment in the young SINE group (right panel). **b** A heatmap for the enrichment of transcription factor binding motifs in SINE families, chromatin accessibility, and histone modifications. **c** The signal density of MeDIP-seq and CpG islands within different SINE families. The shading represents the 95% confidence intervals, shown as error bars above and below the mean column. **d** Boxplots display the DNA methylation levels on different SINE families. "L" and "R" represent the upstream and downstream directions of the SINE body. For example, "L250" represents the 0 to 250 bp window upstream of SINE, and "L500" represents the 250 to 500 bp window upstream of SINE. The line inside each boxplot represents the median.

estimated the contribution of TEs to gene expression across 52 tissues and 27 cell types by analyzing 3570 published RNA-seq samples from the EBI database (Fig. 4a, Supplementary Data 3).

After processing the raw data using LoRDEC[82], we obtained 30,331,870 error-corrected Iso-seq reads with an average length of 2797 bp. Of these, 7.48% (2,267,973) were TE-related transcripts (the transcripts containing TE sequences)[83]. Notably, 68.81% (1,560,568) of the TE-derived transcripts were recognized as SINE-associated transcripts inserted by nearly full-length SINE (average coverage of 87.76%). This suggests that TE-derived transcripts, particularly SINEs, are abundant in the pig transcriptome. We next classified 337,746 young (younger and youngest) SINE-associated transcripts into four categories based on their genomic location compared to known transcripts in the pig genome annotations[81] (Supplementary Fig. 9). Out of these, 1028 perfectly matched with 517 genes and 47 lncRNAs (Supplementary Data 4), while 62,304 potentially offered novel alternative splice events for 8103 genes and 405 lncRNAs (overlapping with at least one splice junction of a known transcript). The remaining young SINE-associated transcripts were classified as either 150,469 exon-covered transcripts (Overlapping exons without splice junction on same or opposite strands) or 130,180 intronic transcripts (Located in an intron), which had no complete structural similarity with the available transcript annotation.

When comparing the locations of young SINEs in their derived transcripts, we noticed a higher proportion of them in UTRs

(Fig. 4b and Supplementary Data 5). Especially, for SINE-associated annotated transcripts that perfectly matched, 81.52% were found to have SINE in their 3′-UTRs. This may affect gene expression by increasing the length of the UTRs[84] or by directly inserting into the regulatory region via a mechanism similar to Staufen-mediated decay (SMD)[85]. For example, we found a full-length PRE0-SS was inserted in the 3′-UTR of the pig *PDK1* gene, which is consistent with a previous report that Alu and B1 regulate both human and mouse orthologs of *PDK1* by SMD[86]. Besides, we found that young SINEs that produced transcripts (Young-T) had lower average CG methylation levels than all young SINEs in most tissues (Fig. 4c). Similarly, these young SINEs were more enriched in open chromatin and histone modifications, particularly H3K4me3, suggesting that they were more likely active across multiple tissues (Fig. 4d).

To study young SINE-associated transcripts, we analyzed transcriptome data from 52 tissues and 27 cell types using Salmon tools[87] to measure gene and SINE-associated transcript abundance (RNA-seq counts). Normalized gene expression by DESeq2[88] allowed us to create t-SNE plots that is mostly consistent with tissue types (Fig. 5a). We also conducted a co-expression network analysis of 14,403 genes using the WGCNA R package[89] to investigate the expression characteristics of young SINE-associated transcripts across a wide range of tissues and cell types (Supplementary Fig. 10).

As a result, 13,872 genes were grouped into 40 modules (number of genes >30), with most modules showing high tissue

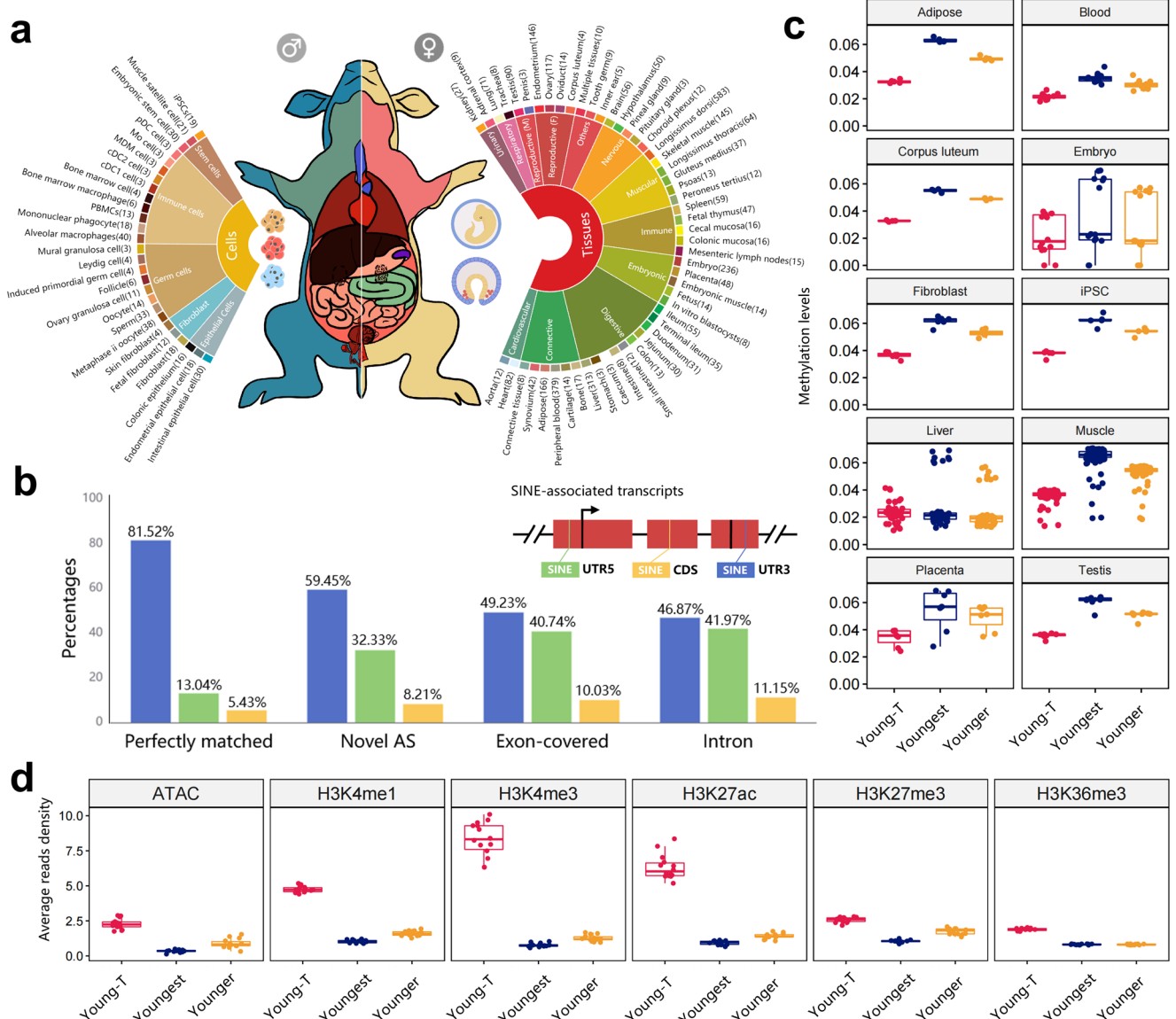

**Fig. 4 Young SINE-associated transcriptome landscape. a** Overview of RNA-seq libraries in 3570 samples across 52 tissues and 27 types of cells. **b** The bar plot indicates the proportion of functional regions affected by SINE across four different categories of SINE-associated transcripts. The x-axis groups indicate the relationship between the SINE-associated transcript and annotated genes, while the colors represent the position of the SINE within the SINE-associated transcript. **c** Boxplots display the CG methylation levels on young SINE families. Young-T group represents the SINE families that derived the young SINE-associated transcripts. The Younger and Youngest groups represent all the younger and youngest SINEs in the entire genome, respectively. **d** The boxplots display the read density of chromatin accessibility and histone modifications in the Young-T, Younger, and Youngest groups. The line inside each boxplot represents the median.

specificity and playing key roles in particular organ systems in pigs (Supplementary Figs. 11, 13, Supplementary Data 6). The results were supported by gene-to-gene networks of topological clustering using the Markov clustering (MCL) algorithm[90] (Supplementary Fig. 14). Importantly, 17.9% of co-expressed genes (2744) were related to young SINE-associated transcripts and were mainly found in specific tissue modules (trachea, adipose, fetal thymus, and alveolar macrophages) (Supplementary Fig. 15).

Young SINE-related genes, as shown in Fig. 5b, were found to be highly enriched in neural development, cellular metabolism, muscle development, and immune response, which may have played a role in natural selection and domestication of modern pigs[91–93]. For instance, 186 genes in module ME2 had high expression in brain tissues and were significantly enriched in

chemical synaptic transmission (GO:0007268), brain development (GO:0007420), and neuron projection morphogenesis (GO:0048812) (Supplementary Fig. 16). Correspondingly, there were 248 young SINEs that were associated with these 186 genes. These SINEs were more enriched in active epigenetic marks (H3K4me1, H3K4me3, and H3K27ac) and depleted from the repressive mark (H3K27me3) at the nervous system (cerebellum, cortex, and hypothalamus) than other tissues, indicating that young SINEs exhibited strong and concordant tissue specificity in both transcript expression and epigenetic regulation (Fig. 5c).

**The roles of young SINEs in population admixture and local adaptation in pigs.** Since the majority of TE polymorphisms in the pig genome are young SINEs[47,55,57,94], we created a thorough map of dimorphic SINEs using whole-genome sequencing data

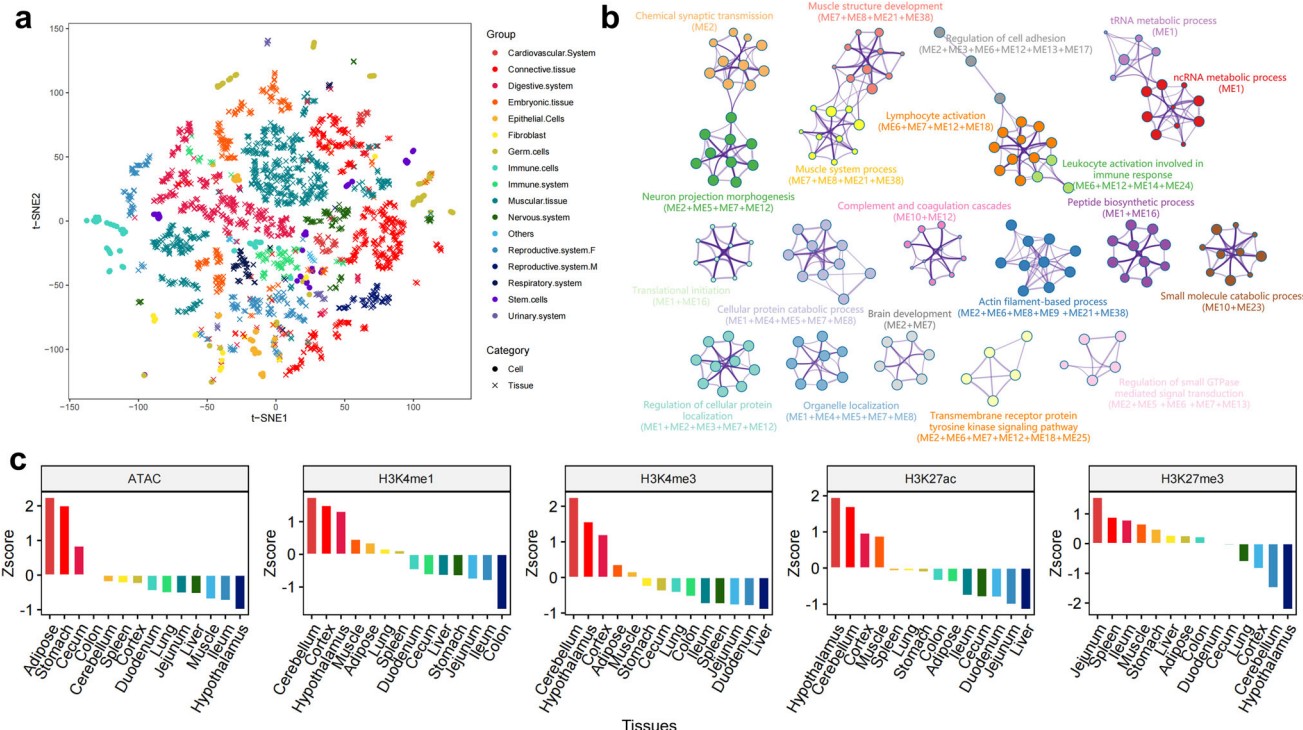

**Fig. 5 Functional enrichment of young SINE-associated transcripts. a** The t-SNE plots display the expression differentiation among different tissues and cells. **b** Top 20 results of functional enrichment analysis for young SINE-associated genes. **c** The bar plot indicates the enrichment of 248 young-T SINEs in chromatin states and histone modifications across different tissues.

from over 300 pigs, which represent most Eurasian pig breeds. This map allows us to investigate the roles of young SINEs and their associated genes in pig population admixture and local adaptation.

To investigate the effect of sequencing depth and dimorphic SINE detection tools on the detection of dimorphic SINEs, we benchmarked polymorphic TE detection tools that previously performed well in human projects under different sequencing depths[13,95] (Supplementary Figs. 17–25, Supplementary methods) and customized the Pig TE Polymorphism (PigTEP) pipeline to improve its performance in pig whole-genome sequence datasets (Fig. 6a, Supplementary methods). We then used this pipeline to identify dimorphic SINEs in 374 individuals from 25 diverse populations ($N \geq 5$) using a uniform sequencing depth of 10× (Supplementary Fig. 26, Supplementary Data 7, average mapped bases: 27.18 GB and average mapping rates: 99.44%). These individuals were assigned to 10 major groups: PYGMY, ISEA, CHD, KOD, AWB, TWB, EUD, EWB, MINI, and COM (Fig. 6b).

We identified a total of 211,067 dimorphic SINEs in the pig genome, with almost half located in non-intergenic regions. Out of these, 189,966 Ref+ refer to the insertion of a TE into the reference genome, while 21,101 Ref- refer to the deletion of a TE from the reference genome. Most of these SINEs (64.89%) were rare, with minor allele frequencies of less than 5% in the entire pig population (Supplementary Fig. 27), and they showed variable frequency distribution among groups (Supplementary Fig. 28). While over 85% of the dimorphic SINEs were shared among groups, there were still 30,441 dimorphic SINEs (PYGMY and ISEA accounted for 60.83% and 26.01%, respectively) that were unique to a single group (Fig. 6c).

Principal component analysis (PCA) analysis of dimorphic SINE genotypes distinguished four species of the Suidae (Fig. 6d). PC1 separated *Porcula slavania* from *Sus* species, while PC2 and PC3 (19.41% and 13.67%, respectively) showed genetic separation between Asian and Western breeds (Fig. 6e). Korean domestic

pigs (KOD) showed closer genetic similarity to Western breeds than to Chinese breeds, likely due to gene flow and introgression mediated by humans. Our results were confirmed by the TE-based phylogenetic tree and genetic admixture (Fig. 6f, g; Supplementary Fig. 29), consistent with previous studies on SNP-based genotypes[44,96]. The comparison of Chinese and European domestic pigs confirmed our previous findings on TE-based introgression between Northern Chinese domestic pigs and European domestic pigs[55]. Korean wild boars clustered with other Asian wild boars instead of European pigs, unlike Korean domestic pigs.

To identify dimorphic SINEs linked to local adaptation, we chose the 10 ancestral components with the lowest coefficient of variation (CV = 0.229; Fig. 6g). For each dimorphic SINE between cluster i and the other clusters, we computed pairwise $Fst_i$ values and alpha coefficients (using Bayescan)[97] to measure their divergence in allele frequencies at specific loci. Loci with higher $Fst$ values and positive alpha values indicate positive selection. We found 337 dimorphic SINEs with high $Fst_i$ and positive alpha coefficient in the gene functional regions, including exon, splice, UTR5, UTR3, and upstream regions, of 330 genes (Supplementary Data 8). 77.94% of the genes were found in both PYGMY ($n = 223$) and ISEA ($n = 42$), while the remaining 75 were linked to breed-specific traits of domestic pigs (Fig. 7a). For example, the PRE1 insertion in the promoter of the *IGFBP7* gene, which is associated with tumor suppression[98], was more common in Chinese indigenous breeds than in commercial breeds[99]. Furthermore, the upstream of *FRZB* (high signals in H3K4me1 and H3K27ac, Fig. 7b), which is associated with pig growth traits[100], was inserted by a population-specific dimorphic SINE from Southern Chinese domestic pigs.

A fixed dimorphic SINE was found in the first exon of the *RUNX3* gene of Goettingen miniature pigs, MiniLEWE, and Southern Chinese domestic pigs, particularly Luchuan pigs (Fig. 7c). *RUNX3* gene is known as a tumor suppressor gene in

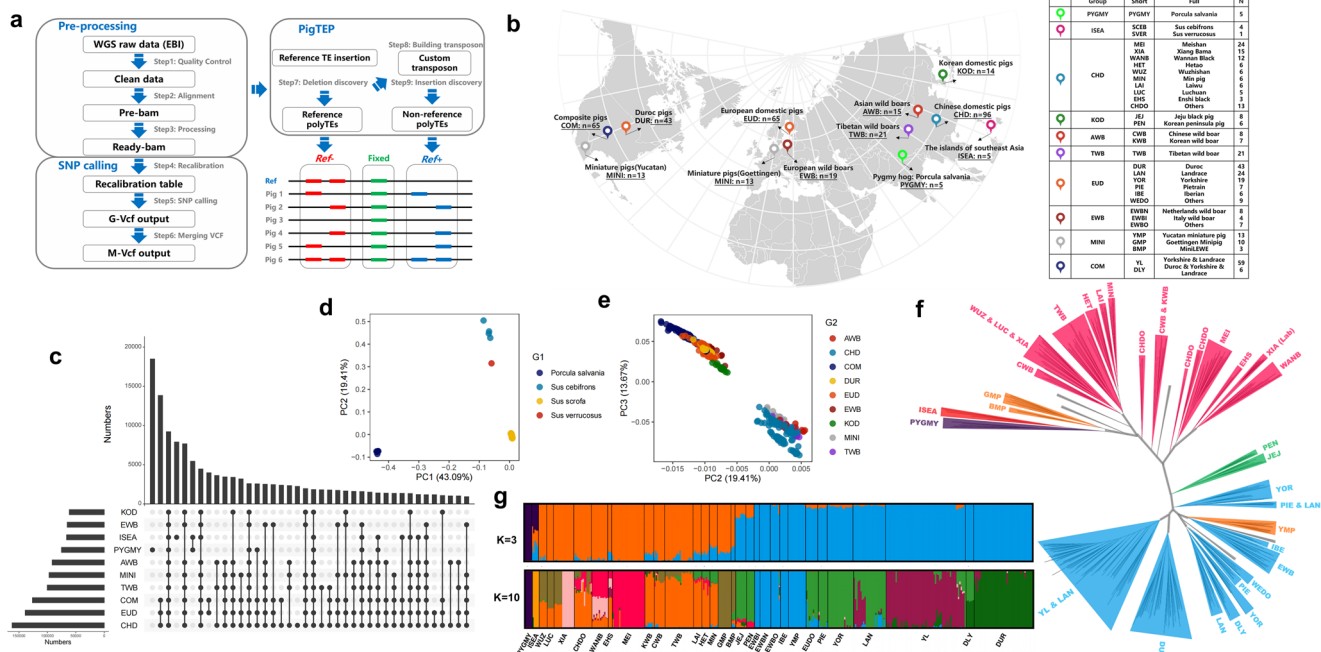

**Fig. 6 Young SINE-associated genetic diversity of pigs. a** The Pig TE polymorphism pipeline. The pipeline was constructed to identify both dimorphic SINEs and SNPs for each individual simultaneously. *Ref+* refers to the insertion of a TE into the reference genome, whereas *Ref-* refers to the deletion of a TE from the reference genome. **b** Overviews of whole-genome re-sequencings in 374 individuals. **c** Venn diagram represents the distribution of dimorphic SINEs among different populations. **d** PCA plot displays the genetic relationship based on dimorphic SINEs among 374 individuals. **e** PCA plot displays the genetic relationship based on dimorphic SINEs among 364 individuals from modern pigs. **f** Phylogenetic tree based on dimorphic SINEs for 374 individuals. **g** Population structure based on dimorphic SINEs for 374 individuals when K was 3 and 10.

human T-cell malignancy[101] and plays a key role in the TGF-β induced signaling pathway[102]. In addition, 30 genes in long-term laboratory-inbred Bama Xiang pigs[103] showed extreme *Fst* and were significantly enriched in the AMPK signaling pathway (Corrected *P*-values = 0.00295, Supplementary Data 9), with *SLC3A2* (upstream) and *SIRT1* (UTR3) genes having the dimorphic SINEs with a perfectly fixed frequency.

Out of 75 candidate genes, nine were mapped to swine quantitative trait-associated (QTX) data associated with phenotypic traits[104] (Fig. 7d). Four of these genes, related to laboratory-inbred Bama Xiang pigs, were associated with fat content and body weight, consistent with selective breeding[103]. The *LEP* gene, highly expressed in adipose tissue, produces leptin, a hormone regulating appetite, fat storage, and body weight[105]. These findings demonstrate that dimorphic SINEs are a valuable source for studying genomic ancestry and local adaptive evolution in pigs.

**Mapping young SINEs to the genetic associations of complex traits.** To investigate the relationship between dimorphic SINEs and complex traits, we collected 4072 trait-associated SNPs (T-SNPs) from 79 published GWAS studies of 97 complex traits in pigs, including reproduction, production, meat and carcass, health, and exterior traits (Supplementary Fig. 30). As shown in Fig. 7e, 127 dimorphic SINEs associated with traits and in linkage disequilibrium (LD, $r^2 > 0.3$) with T-SNPs were identified using 296 domestic pigs (109 Asian and 187 European). Specially, it was found that these dimorphic SINEs were more prevalent in the TxFlnkWk (Weak transcribed at gene), indicating their potential for gene regulation (Fig. 8a).

54 genes influenced by dimorphic SINEs were linked to intramuscular fat composition and teat number. Many of these genes were specifically expressed in certain tissues (Z-score >2), such as the nervous system (plasmacytoid dendritic cells, choroid

plexus, hypothalamus, and brain), reproductive system (testis, oviduct, and oocyte), and muscle satellite cells (Supplementary Fig. 31, Supplementary Data 10). Importantly, most of the intronic T-dimorphic SINEs exhibited breed-specific MAF between Chinese and Western pigs, which is consistent with their differences in fatty acid content and teat number (Fig. 8b).

We found a 320 kb dimorphic SINE hotspot (chr14:112,965,840–113,285,513; $r^2 > 0.3$) linked to intramuscular fat composition, containing six dimorphic SINEs and eight genes. Two genes, C14H10orf76 (*ARMH3*, $r^2 = 0.89$) and *GBF1* ($r^2 = 0.86$), are essential for Golgi maintenance and secretion[106]. The *ELOVL3* gene is a strong candidate gene for fatty acid composition[107,108]. A low-frequency dimorphic SINE was found in its intron region, while multiple T-dimorphic SINEs were found within its upstream region of 15 to 50 kb. The dimorphic SINE (chr14:113,199,425) near 27 kb upstream was found at high frequency in Chinese domestic pigs, especially Southern Chinese domestic pigs (Fig. 8c).

Furthermore, pairwise LD ($r^2 = 0.88$) was observed between the dimorphic SINE (chr8:109,447,835) located in the *ANK2* intron and the T-SNP linked to C14:0, C16:0, and C16:1n7 fatty acid content in backfat[109]. Ankyrin-B (AnkB), an alternatively spliced variant of *ANK2*, is linked to obesity susceptibility in humans[110]. We found that the insertion of the T-dimorphic SINE was almost fixed in Western domestic pig populations (Fig. 8d) and located in an LD block of 15 kb ($r^2 > 0.5$, chr8:109,439,023–109,454,866, Supplementary Fig. 32). We observed that *ANK2* is a gene that is ubiquitously expressed in pigs and highly enriched in the nervous system (Fig. 8e). Two SINE-associated transcripts overlapped with *ANK2* exons and were significantly correlated with *ANK2* expression (Supplementary Data 11). *ANK2* expression was significantly upregulated in cultivars with high-fat deposition such as Songliao black pigs compared with those with low-fat deposition like Landrace

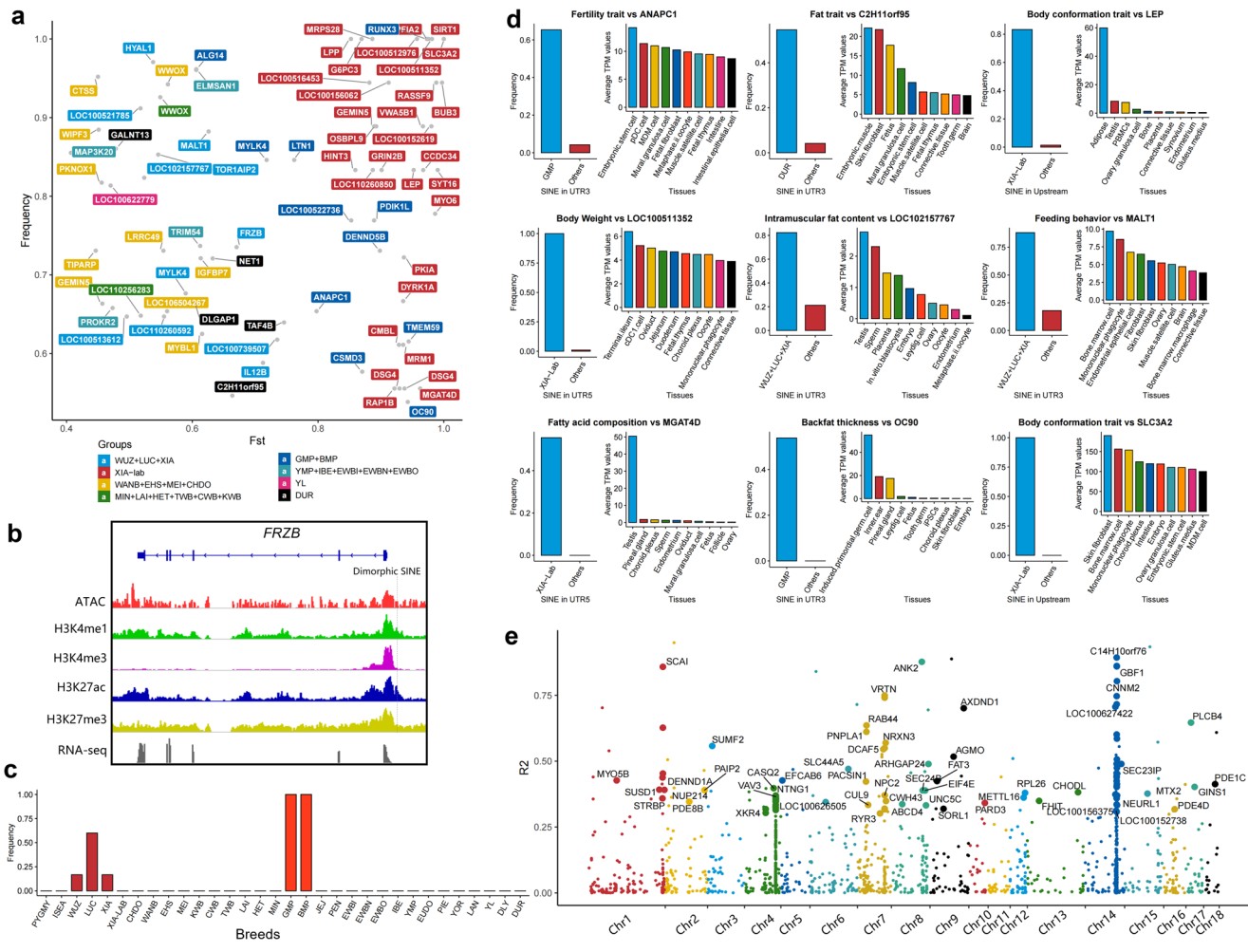

**Fig. 7 Potential candidate genes for young SINE-associated local adaptation. a** The scatter diagram displays the 75 genes that may be associated with local adaptation. The x-axis represents the *Fst*, and the y-axis represents population frequency. **b** Chromatin accessibility and histone modifications for *FRZB* (chr15:88332856–88377275) and its dimorphic SINE (chr15:88377793). **c** Bar plot displays the population frequency of the dimorphic SINEs in the first exon of *RUNX3*. **d** Overviews of nine candidate genes under local adaptation. Bar charts indicate the population frequency of candidate dimorphic SINEs and the average TPM values of their corresponding candidate genes across tissues. **e** The scatter diagram displays the linkage disequilibrium between T-SNPs and dimorphic SINEs. The x-axis represents the chromosome, and the y-axis represents the $r^2$ values.

pigs[111] and fast-growing chickens compared with slow-growing chickens[112].

In addition, we found a high LD ($r^2 = 0.75$) between a T-SNP (chr7:97,606,621) and a T-dimorphic SINE (chr7:97,615,896) in the first intron of the *VRTN* gene—the gene suggested to be associated with teat number and the most promising candidate gene to increase the number of thoracic vertebrae (ribs) in pigs[113]. We observed a clear difference in frequency of the dimorphic SINE between Chinese indigenous breeds and commercial breeds (Fig. 8f). In addition, *VRTN* gene is highly expressed in embryonic stem cells, embryos, and ovaries, suggesting its role in early pig development (Fig. 8g). Especially, a novel transcript (2191 bp in length) derived from the bimorphic SINE covers the first exon of *VRTN* and is significantly associated with *VRTN* expression (Fig. 8h, *P*-values = 3.03 × 10^{-201}, Supplementary Data 11). This transcript was supported by the RNA-seq exon coverage in NCBI annotation (Supplementary Fig. 33). This region exhibited the open chromatin and enhancer signals (H3K4me1) while was facultatively repressed in most tissues (H3K27me3) (Fig. 8i). An obvious decline was observed in the repressed states of stem cells and embryo-related tissues, corresponding to the tissue-specific expression in *VRTN*,

suggesting that this region was crucial for *VRTN*, and this dimorphic SINE was more likely to affect its expression.

## Discussion

In this study, we built a pig genome atlas of TEs using the Pig-TEDC pipeline, combining the similarity-, structure-, and de novo-based methods. Our findings show that almost a third (947 MB) of the pig genome is made up of TEs, mainly non-LTR retrotransposons (SINE and LINE). SINE is shorter and more complete than LINE. Similar to our previous findings[52], the PRE1-SS, PRE0-SS, and PRE1a families in SINE/tRNA are the most recent and have the most polymorphic insertions. These polymorphic SINEs contribute nearly 90% of medium-length SVs across different assemblies, especially the L13 subfamily, with 36.15% of the youngest SINEs in the pig genome.

Gene regulatory network is influenced by genomic components, chromatin accessibility, histone modifications, DNA methylation, and cis-regulatory elements such as TFBS, promoters, and enhancers. TEs linked to specific chromosome features can impact gene regulatory networks in multiple ways as listed above. To our knowledge, this is the first time that large-scale multi-omics data were used to fully explore the relationships

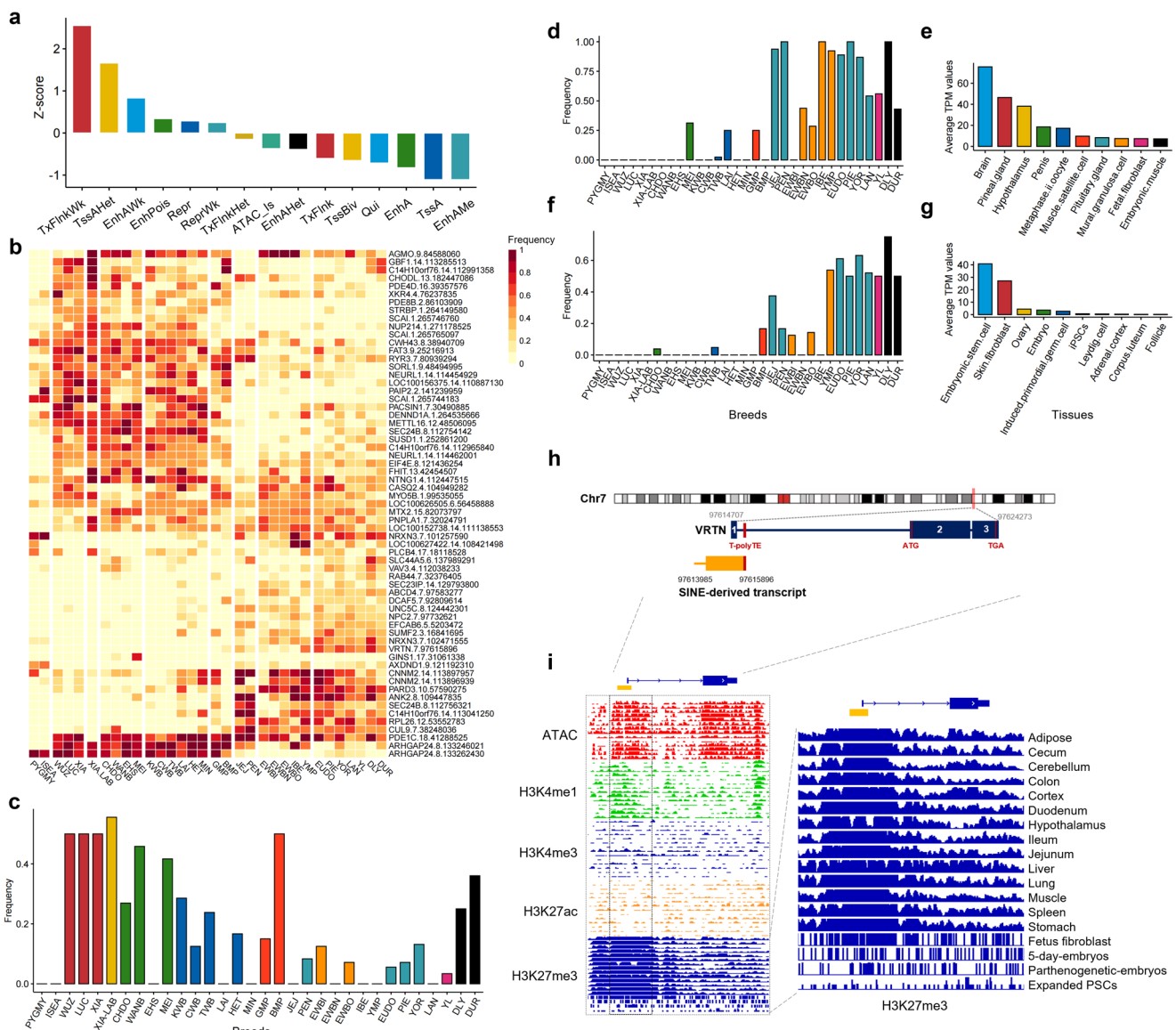

**Fig. 8 Mapping young SINEs to the complex traits. a** Bar plot displays the enrichment of dimorphic SINEs in different chromatin states. **b** Heatmap displays the frequency of T-dimorphic SINEs among different pig populations. The darker red color represents a higher population frequency for dimorphic SINEs. **c** The population frequency of dimorphic SINEs in *ELOVL3* gene among different pig populations. **d** The population frequency of dimorphic SINEs in *ANK2* gene among different pig populations. **e** The expression of the *ANK2* gene in the top 10 tissues sorted by gene expression. **f** The population frequency of dimorphic SINEs in *VRTN* gene among different pig populations. **g** The expression of the *VRTN* gene at the top 10 tissues sorted by gene expression. **h** The *VRTN* gene (chr7:97614707–97624273) structure and the neighboring SINE-associated transcript (chr7:97613985–97615896). **i** Chromatin accessibility and histone modifications for the upstream of *VRTN* gene. H3K27me3 signals for the upstream of *VRTN* gene.

between TEs and chromosome features in the pig genome. Our findings showed that SINEs were highly enriched in the A compartment, and that the enrichment of SINEs in chromatin was associated with their ages. For instance, young SINE families were frequently enriched in close chromatin-like nucleosomes but highly depleted from open chromatin.

As expected, SINEs were highly depleted from all active chromatin tags, and more signals of constitutive heterochromatin tags (H3K9me3 peaks) were observed on SINEs. The exception was H3K27me3, which was associated with facultative suppressor genes and cannot permanently silence SINEs. Most histone modifications in SINE decreased as the TE's age increased, which was in line with the distribution of DNA methylation on SINE and its contribution to TFBS. However, young SINEs, especially the youngest SINE family, were highly enriched in weakly active enhancer regions of hypothalamus tissue (Fold >1.5). We

speculate that the relationship between SINE and its host genome is a combination of both arms race and co-evolution, depending on how the symbiosis turned out.

In the former case of parasitism, the young SINEs were more likely to be treated as new invaders that were constitutively silenced by histone modifications and DNA methylation of the host genome (e.g., PIWI-piRNA pathway during the TE in testis), while the old SINEs mutated and gained new regulatory potential, and thus were tolerated or even co-opted by the pig genome. In the latter case of mutualism, there might be rare cases where the SINEs were positively selected by nature, thereby helping the host genome better adapt to the local environment in the long run.

The use of long-read isoform sequencing provided us a more complete characterization of full-length transcripts, which made it possible to identify the young SINE-associated transcripts. Meanwhile, the Iso-seq reads we used here were collected from

~40 pig tissues, which ensured the investigation of the abundance and tissue specificity of young SINE-associated transcripts.

Our findings showed that the vast majority of young SINE-associated transcripts were non-coding RNAs that covered exons or fell within introns. A total of 3112 genes were found to be associated with young SINE-associated transcripts, and nearly 88% of them were enriched in co-expressed modules with high tissue specificity. The transcripts derived from young SINEs exhibited lower levels of CG methylation and were more enriched in open chromatin and histone modifications than whole young SINEs. Specifically, some young SINEs exhibited strong and consistent tissue specificity in both transcript expression and epigenetic regulation. This is consistent with previous findings in other species[114–116], suggesting that SINE insertions may be a crucial component of genes and regulate tissue-specific expression of their target genes.

Dimorphic SINEs belong to SVs that are more sensitive to sequencing depth than SNPs. In this study, we benchmarked four dimorphic SINE detection tools under different sequencing depths to ensure the unbiased detection of dimorphic SINEs. Our findings showed that MELT (version 2.2.2) had robust performance in both *Ref* + and *Ref*- detection (Supplementary Figs. 17–18). As expected, we found that the number of detected dimorphic SINEs increased with sequencing depth, especially from 5× to 10×, which nearly doubled the average number of dimorphic SINEs (Supplementary Fig. 19). Considering the sequencing depth in the current 838 publicly available whole-genome sequence datasets in pigs, we retained 374 individuals whose sequencing depth was greater than 10× and down-sampled their sequencing depth to ~10× through a strategy of randomly removing reads (average mapped bases: 27.17 GB and average mapping rates: 99.43%). Finally, the PigTEP pipeline was developed to identify both dimorphic SINEs and SNPs in individuals simultaneously. This pipeline will help other researchers explore the role of dimorphic SINEs in pig genomic study and breeding.

The contribution of TEs was underestimated in pig genomic research, despite the active role of SINEs under selective pressure. However, our understanding of SINEs in pig population genetics is limited without a comprehensive map of dimorphic SINEs based on large-scale re-sequencing data.

We genotyped and analyzed 211,067 dimorphic SINE loci in 374 individuals from 25 pig populations. These loci showed high variability in allele frequencies among populations. Based on these SINEs, we identified ten major clusters that corresponded to geographic differentiation. These SINEs with high pairwise *Fst* value can help us understand local adaptation in domestic pigs and identify candidate genes for economically important traits. Our findings confirm previous studies (e.g., *IGFBP7*) and identify new candidate genes.

GWAS studies have discovered thousands of QTLs for important pig traits based on SNPs, but most of these loci remain functionally uncharacterized. One possible reason is that phenotypic changes may be affected by SVs (TEs) in linkage disequilibrium with SNPs. In this study, 127 dimorphic SINEs were found to be in linkage disequilibrium with significant GWAS SNPs of complex traits, and nearly a third of them showed high tissue specificity in expression. Some of these dimorphic SINEs can generate novel transcripts (in H3K4me1 and H3K27me3), as exon-covered transcripts were found upstream of the *VRTN* gene. Future research is needed to validate how these dimorphic SINEs regulate target genes in specific tissues (e.g., *VRTN* gene in embryonic stem cells) and affect complex traits.

## Methods

**Overview of the PigTEDC pipeline**. The PigTEDC pipeline was composed of three TE detection approaches using different algorithms to achieve its overall efficiency:

The similarity-based method was represented by the widely known RepeatMasker (V4.0.6) (http://www.repeatmasker.org), which search sequence using the TE consensus sequence from RepBase Update (https://www.girinst.org/repbase/) and Dfam 2.0 databases[117] against homologous regions of the pig genome. The method was used to detect all known TEs, including DNA transposons, LTR, SINE, and LINE.

The structure-based method was used to capture the particular TE families based on their known sequence structure and motifs, which further enhanced the results of the similarity-based method to enhance the power of detecting known TEs. In our pipeline, HelitronScanner (V1.1)[118] was used to improve DNA transposons detection, both LTRharvest (V2.0)[119] and LTR-Finder (V1.07)[120] were used to enrich the LTR results, and SINE-Finder (V1.1)[121] was used to increase SINE detection.

The de novo-based method was used to identify the missing pig TEs from the known TE database, which employed the clusters of the repetitive sequences in the genome based on various methods. There were a total of three tools in this category. RepeatModeler (V1.0.11) was used to automate the runs of RECON (Multiple alignment clustering) and RepeatScout (Consensus seed clustering) for the pig genome (http://www.repeatmasker.org/ RepeatModeler/). Red[122] and P-Clouds[123] were used to capture all potential repetitive sequences that included TEs and simple repeats in the pig genome using machine learning and oligonucleotide clustering methods, respectively.

The classification of pig TEs detected in the PigTEDC pipeline was further performed by the RepeatMasker through the following three parts:

Order and family classification of known TEs. The known TEs generated from both similarity-based and structure-based methods can be directly classified into four orders by referring to the RepBase update database, including DNA transposons, LTR, SINE, and LINE. The results of RepeatModeler from the de novo-based method also provided parts of known TEs as above. Considering the complex structure of nested TEs, we removed these TEs from RepeatMasker using parseRM_GetNesting.pl. After merging known TEs with each order, we removed the redundant TEs detected by different tools by using bedtools. For example, some SINEs can be identified by both RepeatMasker and SINE-Finder. At last, the non-redundant known TEs from each order were aligned to the consensus sequences of various families belonging to that order to confirm their specific families.

Match of known TEs with their family consensus sequences. Considering the different TEs have different coverage towards their family consensus sequences, we located the coverage area for each TE family using cross_match software (http://www.phrap. org/consed/). The parameters of cross_match was set as "-gap_init −25 -gap_ext −5 -minscore 10 -minmatch 6 -alignments -bandwidth 50 -word_raw". Parts of TEs failed to pass the paired match with their family consensus sequences, thus were regarded as overly fragmented TEs. The passed TEs consisting of full-length and fragmented TEs, had clear and specific family classifications.

Family classification of unknown or novel TEs. In the step of classifying unknown Pig TEs, we first merged all repetitive sequences from both Red and P-Clouds tools, and kept parts of them with at least three copies using hs-blastn. We then filtered these multicopy sequences to keep potential TE sequences by aligning them with various non-TEs repetitive sequences, including tandem repeat, gene, tRNA, and rRNA. Meanwhile, these potential TE sequences were additionally filtered as unknown TEs by excluding known TE sequences from the RepBase update database. In addition, merging all unknown TEs from RepeatMasker, RepeatModeler, and Red & P-Clouds, we further reduced the presence of redundant sequences by CD-HIT

(parameters: -c 0.90 and –n 8) and picked the representative one of each cluster as the consensus sequence for each unknown TE family. Finally, to obtain the genomic locations of unknown TEs, we performed the genome-wide identification for each unknown TE family using RepeatMasker with the custom-build library.

**Divergence distribution of pig TEs**. Divergences of TEs from consensus sequences were obtained using the Kimura two-parameter model from RepeatMasker[124]. The divergence levels were corrected for CpG content by $DCpG = D/(1 + 9FCpG)$[125]. Histograms with a bin size of 0.01 were plotted to show the distribution of divergence levels. Activity periods were estimated assuming a substitution rate of $5 \times 10^{-9}$ substitutions/site per year[126,127]. Distribution histograms for sequence divergence of TE were plotted using a 0.01 bin size.

**Processing of young SINEs**. Annotation of young SINEs on 14 genome assemblies. For each pig genome assembly, we employed RepeatMasker to search for young SINEs in our customized repeat library (PRE1-SS, PRE0-SS, and PRE1a). And then "parseRM_GetNesting.pl" was used to remove the nesting and nested young SINEs, followed by a length filtering to remove elements with length <200 bp and >300 bp using a shell script.

Clustering and filtering of young SINEs. To ensure that the young SINEs used for subfamily classification were highly homologous, we utilized a clustering-based approach to keep all non-nested young SINEs from 14 genomes with a sequence identity of >90%, implemented in CD-HIT-EST[128] with the parameter that "-T 0 -c 0.9 -M 0 -n 5 -p 0". we finally retained 52 of 17,294 SINE clusters ($n > 500$) involving 1,157,133 young SINEs.

Removing the duplicated sequences of young SINEs. Due to the existence of shared SINEs from different pig genomes, we removed the completely duplicated sequences from our young SINE datasets using seqkit (https://github.com/shenwei356/seqkit).

Construction of consensus sequence. Because the running time and memory usage for multiple sequence alignment (MSA) of large-scale genomic sequences can be enormous, we randomly extracted 100,000 young SINEs to conduct the MSA. Using MEME suite programs (http://web.mit.edu/meme_v4.11.4/share/doc/motif-consensus.html), we scanned each column in a letter's (A, T, C, G) frequency matrix of MSA using the "50% rule" that any letters with frequency less 50% of the maximum were discarded. Finally, a consensus sequence with a length of 261 bp was created for the subfamily classification.

**Phylogenetic tree for SINEs**. 26 SINE families were used to construct the phylogenetic tree. Multiple sequence alignments of their consensus sequences were performed with mafft[129] (V7.407) at the default setting. Then, IQ-TREE[130] was used to create maximum likelihood (ML) trees for SINE (with 100 fast bootstrap replicates). Finally, EVOLVIEW[131] was used to visualize the phylogenetic tree for SINE families.

**Subfamily classification of young SINEs**. We conducted subfamily classification for young SINEs using the COSEG pipeline (http://www.repeatmasker.org/COSEGDownload.html). First, sequence homology analysis was done using cross_match software (https://www.phrap.com/) with the parameters (-gap_init −25 -gap_ext −5 -minscore 100 -minmatch 6 -alignments -bandwidth 50 -word_raw). Then, "preprocessAlignments.pl" was used to determine the consensus range and create input files for COSEG programs with the following parameters: the minimum distance between sites (-maxEdgeGap) was 10; the consensus

sequence ranged from 1 to 251 bp (-minConsRange 1 -maxConsRange 251; refer to Supplementary Fig. 1). In the end, a minimum spanning tree of young SINEs was constructed to define their subfamilies using COSEG programs with the following parameters: the minimum subfamily size (-m) was 100; Alkes Price's $p$-value method (-k) was used[132]; 2 bp (-t) co-segregating mutations were used when developing subfamilies.

**Identification of dimorphic SINEs among the 14 genome assemblies**.

(1) Detection of SVs among the 14 genome assemblies. We employed Minimap2 (V2.17)[133] to identify SVs in the assemblies compared to the pig reference assembly. First, we performed a cross-breed full-genome alignment (PAF file) for each genome assembly using Minimap2 with the following parameters: "-c -x asm20 –cs". Then, we used paftools.js (a JavaScript script within Minimap2) to identify the confident/callable regions and call the variants from the asm-to-ref alignment with the following parameters: "call -L1000".

(2) Identification of dimorphic SINEs from SVs. We first retained the SVs that had a similar size (>200 bp and <300 bp) with overlapping SINE using a shell script. Then, the inserted sequences of the resulting SVs were then aligned against the consensus sequences of SINE families and young SINE subfamilies using BLASTn version 2.2.31+ with default parameters. Finally, the SINE families or subfamilies of these SVs were defined by their best high-quality match between consensus sequences of SINE families or subfamilies.

**3D chromatin architecture and chromatin accessibility**. The 3D chromatin architecture and ATAC-seq used in this study, including A/B compartments, Hi-C, and TADs, were downloaded from the FRAGENCODE project[62]. MNase-seq reads were filtered with Trim_galore tools to obtain and retain high-quality reads with the following parameters: "-q 20 --phred 33 --stringency 3 -e 0.1". The filtered reads were then aligned to the Duroc reference genome (*Sus scrofa* 11.1) using the BWA[134]. DANPOS2[135] (V2.26) was then used to call nucleosome binding peaks and to generate nucleosome occupancy profiles with the following parameters: "--span 1 --smooth_with 20 --wideth 40", which were further normalized with the mean score of the whole genome.

**Histone modifications**. The histone modifications used in this study include four active epigenetic marks (H3K4me1, H3K4me3, H3K27ac, and H3K36me3) and two repressive marks (H3K9me3 and H3K27me3). Histone modification data were quality-filtered using Trim_galore tools with the following parameters: "-q 20 --phred33 --stringency 3 -e 0.1". ChIP-seq data were aligned to the Duroc reference genome using BWA. Peaks were called using MACS2[136] (V2.1.1) with the following parameters: "-q 0.05". The computeMatrix module from deepTools[137] (V.3.5.0) was used to transform and compute the corresponding data matrix from the modification signal over a set of SINE regions. The plotProfile module was used to turn the compressed matrix into summary plots.

**Definition of chromatin status**. As described in a previous study[66], we defined 15 distinct chromatin states across 14 tissues and grouped them into the following seven categories: (1) promoters included TssA (Strongly active promoters/transcripts), TssAHet (Flanking active TSS without ATAC), and TssBiv

(Transcribed at gene); (2) TSS-proximal transcribed regions included TxFlnk (Transcribed at gene), TxFlnkWk (Weak transcribed at gene), and TxFlnkHet (Transcribed region without ATAC); (3) enhancers included EnhA (Strong active enhancer), EnhAMe (Medium enhancer with ATAC), EnhAWk (Weak active enhancer), EnhAHet (Active enhancer no ATAC), and EnhPois (Poised enhancer); (4) repressed regions included Repr (Repressed polycomb) and ReprWk (Weak repressed polycomb); (5) quiescent regions; (6) ATAC_Is (ATAC island); (7) TssBiv (Bivalent/poised TSS).

**Transcription factor binding sites**. To identify TFBS genome-wide, we used the FIMO software[138] from the MEME Suite[139] to look for their occurrences in the Duroc reference genome for the 746 Transcription Factors (TFs) cataloged in the Vertebrate 2020 JASPAR database[140]. We set the parameters of fimo software as "$p$-value = 1e-4", "--max-stored-scores 100000000" and "--alpha=1". Among the predicted TFBS, those obtained from the 31 known TF motif were used to profile the resulting density of different SINE families.

**DNA methylation**. MeDIP-Seq data were quality-filtered using Fastp tools[141] with the following parameters: "-q 20 -u 30 -l 30 -w 16". MeDIP-Seq data were aligned to the Duroc reference genome using Hisat2[142]. Duplicate reads were removed from the bam files using Sambamba tools[143]. The filtered bam files were used to identify the MeDIP-enriched regions based on a clustering approach using SICER2[144].

The WGBS data were aligned to the Duroc reference genome using Bismark[145] (V0.23.0) with the parameters that "--ambig-bam". The statistical analysis and visualization of DNA methylation levels on TEs and the whole genome were performed using the methPlot script from BatMeth2[146], and the R language utilizing the "data.table" and ggplot2 packages.

**Identification of small non-coding RNAs**. To study non-coding RNAs in young SINE families, we did the following steps: (1) We merged all raw data and removed adapters with sRNAseqAdapterRemover from TBtools[147]. (2) We used the "collapse_reads_md.pl" script of Mirdeep2[148] (V0.1.2) to remove repetitive sequences. (3) We downloaded known pig piRNAs from piRBase[149] (V3.0) (http://bigdata.ibp.ac.cn/piRBase/) and used Bowtie to map the collapsed reads to them, saving the unaligned reads. (4) We used Mirdeep2's mapper and miRDeep2 to identify candidate microRNAs and their genomic locations based on these unaligned reads. (5) We mapped the remaining reads that did not align to piRNAs or miRNAs to the Duroc reference genome using Mirdeep2's mapper, filtering for a length of 21 nt to identify candidate siRNAs. We excluded any small non-coding RNAs that had multiple hits in the reference genome.

**Identification of SINE-associated transcripts and gene quantification**. Using LoRDEC[82] software with parameters "-k 21 -s 3", we processed the raw data by their corresponding RNA-seq data. After merging the error-corrected Iso-seq reads, we aligned the consensus sequence of young SINE families to it using BLAST software with parameters "-evalue 1e$^{-5}$ -max_target_seqs 1". We identified 2,267,973 candidate TE-derived transcripts containing young TE insertions (transcripts contain more than 80% of the sequence of SINE). As previously studied[81], we categorized young SINE-associated transcripts into four groups by comparing their genomic location to known transcripts in the currently available pig genome annotations. These four groups, as shown in Supplementary Fig. 9, include: (1) perfect match with known annotation; (2) at least one splice junction in common with a known

transcript; (3) exonic overlap without a matched splice junction on the same or opposite strands; (4) located in a reference intron.

To study the cis-functionality of young SINE-associated transcripts, we used Salmon tools[87] to measure the abundance (RNA-seq counts) of genes and SINE-associated transcripts. The RNA-seq counts were transformed to log2-counts per million (log CPM, suitable for linear modeling[150]).

**Construction of co-expression network**. We used 14,403 genes from 3570 samples to explore young SINE-associated transcripts across different tissues and cell types. We analyzed gene expression using DESeq2[88] and built a co-expression network with WGCNA R package[89]. The network was constructed with the following parameters: "corType = pearson, maxBlockSize = 20,000, power = 4, minModuleSize = 30, mergeCutHeight = 0.3". The power was set to 4 (Supplementary Fig. 10). The blockwiseModules function was used to create a correlation network, a cluster tree, and modules. The modules were merged and analyzed using the plotDendroAndColors function.

**KEGG enrichment and Gene ontology analysis**. We analyzed 2744 genes associated with young SINE-associated transcripts using Metascape[151]. We used human gene IDs for enrichment of KEGG pathway and Gene Ontology terms, with the entire human gene list as the gene background after converting all pig gene IDs.

**Benchmarking detection tools and sequencing depth for dimorphic SINE detection**. We benchmarked four types of dimorphic SINEs detection tools that have been reported to have superior performance in the previous projects[13,95,152], including STEAK[153] (http://github.com/applevir/STEAK), TranSurVeyor[154] (https://github.com/Mesh89/TranSurVeyor), MELT[155] (https://melt.igs.umaryland.edu/), and RetroSeq[156] (https://github.com/tk2/RetroSeq). Their performance was evaluated using Sniffles[157] (https://github.com/fritzsedlazeck/Sniffles) with 20× PacBio sequencing from the same individual. We compared the counts and verification rates (the proportion of dimorphic SINEs supported by Sniffles) of identified dimorphic SINEs across a range of sequencing depths (5x, 10x, 15x, 20x, 30x, and 50x) to assess their detection powers (Supplementary Figs. 16–17). The results showed that MELT showed the robust performance on both *Ref*+ and *Ref*- dimorphic SINEs than the other three dimorphic SINEs detection tools. Therefore, the MELT was selected in our PigTEP pipeline. Using the MELT program, we further performed the dimorphic SINEs detection using NGS data of Meishan and Duroc breeds across a range of sequencing depths (5x, 10x, 15x, 20x, 30x, and 50x). We found that the number of dimorphic SINEs increased with the sequencing depth, especially from 5x to 10x that increased the average number of dimorphic SINEs by nearly one-fold (Supplementary Fig. 18). After evaluating the sequencing depth of current 838 publicly available pig next-generation sequence datasets (Downloaded from the NCBI SRA database https://www.ncbi.nlm.nih.gov/sra), we retained 374 individuals whose sequencing depth was greater than 10x for dimorphic SINEs identification (average mapped bases: 27.17 GB and average mapping rates: 99.43%).

**Identification of SNPs and dimorphic SINEs from re-sequencing data**. Preprocessing of next-generation sequencing. The 374 individuals were selected to identify the SNP and dimorphic SINEs, after their sequencing depth was standardized around 10x. We first performed the preprocessing as following steps: Step 1: Quality control. Quality control was conducted for each raw re-sequencing data using the fastp (https://github.com/OpenGene/fastp) with the following parameters: The quality value that a base

was qualified (-q) was 20; The percent of bases were allowed to be unqualified (-u) was 30; The required length of kept reads (-l) was 50; Step 2: Read alignment. Clean reads from all individuals were aligned to the Duroc reference genome (Sus scrofa 11.1) using BWA[134] (https://github.com/lh3/bwa) with the BWA-MEM algorithm; Step 3: Processing the alignment files. The mapped reads were subsequently processed for format conversion (view), position sorting (sort), merging (merge), and statistics (stats) using Samtools[152] (http://samtools.sourceforge.net/). Then, the MarkDuplicates method from the Picard package (https://sourceforge.net/projects/picard/) was used to remove the PCR duplicates that were introduced during library construction.

Genome-wide SNP detection. After preprocessing of the alignments, the DNAseq mode of Sentieon (https://github.com/Sentieon/sentieon-dnaseq) was next used to identify genome-wide SNP in the following steps: Step 4: Indel realignment and Base quality score recalibration (BQSR). The processed alignment bam files were subjected to realign and recalibration with the parameters of "Realigner" and "QualCal"; Step 5: SNP genotype calling. Genotype calling of SNP for each individual was performed using the "Haplotyper" algorithm. Step 6: Merge of VCF files. All 374 g-vcf files were joined together using the "GVCFtyper" algorithm. Then, the newly merged VCF were filtered to retain high-quality SNPs using the vcftools with the following parameters: "--max-missing 0.8 --maf 0.05 --min-alleles 2 --max-alleles 2 --recode".

Genome-wide dimorphic SINEs detection. The MELT tool (https://melt.igs.umaryland.edu/) was used to discover the dimorphic SINEs (Ref+ and Ref-) in the following steps: Step 7: Discovery of deletion. With the processed bam files from Step 3, we used the Deletion-Genotype module of MELT to identify the "Ref-" dimorphic SINEs based on our previously identified TE sets (bed file, $n = 3,087,929$); Step 8: Building retrotransposon. Referring to the results of Step 7, we build three customized reference files for PRE1-SS, PRE0-SS, and PRE1a families using the "BuildTransposonZIP" module; Step 9: Discovery of insertion. For each bam file, we carried out the identification of "Ref+" dimorphic SINEs using four modules of MELT step by step, including "IndivAnalysis", "GroupAnalysis", "Genotype", and "MakeVCF".

All detected dimorphic SINEs and SNPs were processed using gene-based annotations in ANNOVAR[158]. The pig annotation files were downloaded from the NCBI database for the Duroc reference genome (Sus scrofa 11.1). Dimorphic SINEs and SNPs were classified into eight categories based on their genome locations, including exonic regions, splicing sites, intronic regions, 5′ and 3′ untranslated regions (UTRs), upstream and downstream regions, and intergenic regions.

**Principal component analysis**. We filtered identified dimorphic SINEs (Ref+ and Ref-) from 374 pigs using PLINK[159] v1.9 with following parameters: "--maf 0.01 --mind 0.8 --geno 0.8". Next, we performed principal component analysis on the filtered dimorphic SINEs using PLINK v1.9 with the parameters: "--pca". We presented the eigenvectors and eigenvalues for each individual in the PCA biplot using R packages.

**TE-based phylogenetic tree**. To establish evolutionary relationships between individuals, we transformed the dimorphic SINEs dataset (Ref+ and Ref-) into a pseudo-SNPs dataset (A/T/C/G) by randomly replacing data under the condition that individual polymorphism was not changed. The transformed dataset was reduced to 44,192 dimorphic SINEs using PLINK v1.9 with the following parameters: "--maf 0.1 --indep --pairwise 50 10 0.2", based on a linkage disequilibrium threshold of 0.2 and a minor

allele frequency threshold of 0.1. The phylogenetic tree was constructed with 1000 bootstrap replicates using the maximum-likelihood approach implemented in SNPhylo[160] (V1.10.2).

**Analysis of genetic admixture**. We used the program ADMIXTURE[161] (V1.3.0) to analyze population structure in our study. This estimated genetic admixture among different pig breeds using all identified dimorphic SINEs. We tested 14 cases (ranging from K = 2 to 15) to identify genetic clusters for 374 pigs, using default parameters. Results were visualized using StructureSelector[162].

**Identifying the dimorphic SINEs associated with local adaptation**. As shown in Fig. 6g, the optimal number of ancestral components was inferred to be K = 10 with the lowest CV (0.229), resulting in the eleven distinct genetic clusters of G01: PYGMY, G02: ISEA, G03: Southern Chinese domestic pigs (WUZ, LUC, and XIA), G04: laboratory-inbred Bama Xiang pigs, G05: Eastern Chinese domestic pigs (CHDO, WANB, EHS, and MEI), G06: Asian wild boars (KWB, CWB, and TWB) plus Northern Chinese domestic pigs (LAI, HET, and MIN), G07: Goettingen miniature pigs and MiniLEWE[163], G08: Korean domestic pigs (JEJ and PEN) plus European domestic pigs, G09: European wild boars plus IBE and Yucatán miniature pigs, G10: Landrace/Yorkshire crossbreeds (YL)[164], and G11: Duroc pigs.

We calculated the pairwise $Fst_i$ value and estimated alpha coefficient[97] of dimorphic SINEs between cluster i and the remaining clusters to measure their locus-specific divergence in allele frequencies. Pairwise $Fst_i$ values were calculated for each dimorphic SINEs using the vcftools[165] (V0.1.16) with the default parameters. BayeScan program[97] (v2.1) was used to identify putative adaptive dimorphic SINEs. based on different allele frequencies among populations, and we performed it using default settings (prior odds to 10, iterations to 5000, and burn-in to 50,000).

**Linkage disequilibrium analysis**. We collected a total of 4072 trait-associated SNPs (T-SNPs) from 79 published GWAS studies of 97 complex traits. These included 18 reproduction, 22 production, 36 meat and carcass, six health, and two exterior traits. We combined the dimorphic SINEs and 4072 T-SNPs into a new variant dataset from 296 domestic pigs (109 Asian and 187 European domestic pigs). We analyzed the linkage disequilibrium between T-SNPs and dimorphic SINEs using PLINK v1.9 with the parameter "--ld". We only considered dimorphic SINEs with a relative T-SNP distance less than 10 kb and R2 greater than 0.3 as candidate dimorphic SINEs. Linkage disequilibrium analysis for ANK2 gene (chr8:109,439,023–109,454,866) was performed using Haploview software[166]. LDBlockShow[167] (V1.40) tool was used for the linkage disequilibrium analysis of large regions like 320 kb dimorphic SINEs hotspots (chr14:112,965,840–113,285,513).

**Statistics and reproducibility**. The Z-score of each chromatin state for each TE group was calculated using permutation tests in regioneR[168], with the number of permutations set to 1000. For the association analysis of young SINE-associated transcripts and genes, we used R code and generalized linear models (GLM) to model transcript expression levels of each gene. We considered the expression level of young SINE-associated transcripts (within their gene body) and the type of tissues or cells as explanatory variables. We used a Bonferroni significance threshold of $1.42 \times 10^{-6}$ (0.05/35,135) as the standard threshold.

**Reporting summary**. Further information on research design is available in the Nature Portfolio Reporting Summary linked to this article.

## Data availability

All public datasets of pigs used in our analysis had sufficient ethical approval. Five pig MNase-seq samples were downloaded from NCBI (BioProject ID PRJNA417273) representing brain, heart, kidney, liver, and muscle tissues. H3K4me1, H3K4me3, H3K27ac, and H3K27me3 in 14 tissues were downloaded from NCBI under BioProject ID PRJNA762083. In addition, we downloaded one H3K9me3 sample (BioProject ID PRJNA152995), and four H3K36me3 (BioProject ID PRJNA529704) and H3K27me3 (BioProject ID PRJEB31243) samples each. Chromatin states of pigs were analyzed in 14 different tissues, which were downloaded from (http://genome.ucsc.edu/s/zhypan/susScr11_15_state_14_tissues_new). This study used MeDIP-Seq data from 80 samples, which were downloaded from NCBI under BioProject ID PRJNA143661. Whole Genome Bisulfite Sequencing data from 246 samples across 10 tissues were downloaded from 16 studies (Supplementary Data 12). Data from 177 samples across 20 pig small non-coding RNA studies were collected (Supplementary Data 13). Two PacBio long-read isoform sequencing datasets were downloaded from NCBI: BioProject ID PRJCA000349 (pooled 38 samples) and PRJNA351265. Transcriptome data from 52 tissues and 27 types of cells can be found in Supplementary Data 3. The source data for the main figures is available in the online resource of figshare (https://doi.org/10.6084/m9.figshare.23856213.v4)[169].

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

## Acknowledgements

This work is financially supported by the Project of Sanya Yazhou Bay Science and Technology City (SKJC-2020-02-007), the National Natural Science Foundation of China (31941007), the Science and Technology special fund of Hainan Province (ZDYF2022XDNY237), the AFRI grant numbers 2019-67015-29321 and 2021-67015-33409 from the USDA National Institute of Food and Agriculture (NIFA), and the High-performance Computing Platform of YZBSTCACC. We thank Sylvain Foissac from Université de Toulouse and other members associated with the FRAGENCODE project for their contributions to data of chromatin accessibility used in this manuscript.

## Author contributions

P.Z., Z.W., L.F., and G.L. conceived and designed the experiments. P.Z. designed analytical strategy and performed all analysis process. Y.G. assisted the DNA methylation analysis. Z.P. assisted the Histone modifications analysis. L.L. and X.L. assisted the bioinformatics analysis process. L.G., H.Z., X.H., and L.Q. collected and prepared for sequencing data. P.Z., G.L., L.F., Z.W., H.Z., and D.Y. wrote and revised the paper. All authors read and approved the final manuscript.

## Competing interests

The authors declare no competing interests.
