## [Peer Review File · Communications Biology]

Reviewers' comments:

Reviewer #1 (Remarks to the Author):

I have now reviewed the manuscript of Zhao et al. entitled "Building an atlas of transposable elements reveals the extensive roles of young SINE in gene regulation, genetic diversity, and complex traits in pigs". In their article, the authors perform an in-depth analysis of the transposable elements (TEs) distribution and potential functions across a vast number of pigs using a impressive amount of complementary data including 14 genome assemblies, hundreds of WG-resequencing, transcriptomics and epigenomics dataset.

After a thorough description of the overall TE content, the authors focus on the most abundant SINEs elements, which also includes the most evolutionary recent TE families in pigs. Younger SINEs seem associated with increased silencing (using different measures of chromatin accessibility, nucleosome location, and histone marks) while some rare instances seem to escape repression and appear to contribute in diverse fashions to the gene regulatory network. Further analyses connect these TEs with functional polymorphisms indicated by their co-segregation with variants associated with complex traits. Population genetics analyses using the segregating SINES (poly-SINEs) recapitulates known population structure, and some SINEs are also suspected to show signature of local adaptation.

I found the manuscript well written and the succession of experiments easy to follow. As for the relevance of the experiments, I believe that some aspects of several analyses (as described in the supplementary data) need to be addressed. My general comment is that though general trends can be legitimately drawn from the sum of all the analyses, some individual results may require improved statistical evaluation, or further justification from the authors.

Main:

1 - "Definition of chromatin status and calculation of fold enrichment". An important part of these results rely on the differential enrichment of the SINEs in different functional compartments of the genome according to their relative age group. While the authors present the fold-enrichment for each functional region in each of the 14 considered tissue, each enrichment does not appear to be formally tested. In other words, do the observed enrichments calculated in each tissue deviate significantly from the expectation if TE and functional regions were distributed randomly? This could be addressed by shuffling TE positions across the genome. You can for example use this script: https://github.com/4ureliek/TEanalysis/blob/master/TE-analysis_Shuffle_bed.pl (you can access the full description by calling the help or reading in the code directly. Note that I am not the author of this script).

2 - "Pre-processing of young SINEs"

2.1 - It was unclear how nested SINE copies were dealt with. Starting line 70 of the supplementary data, "'parseRM_GetNesting.pl' was used to remove the nesting and nested young SINEs, followed by a length filtering to remove elements with length > 200 bp and < 300bp using a shell script.": did instances of SINE with an embedded TE, or a SINE within another element were removed? Why not using the nested SINEs?

2.2 line 73: "52 of 17,294 SINE clusters (n > 500) involving 1,157,133 young SINEs" -- In table S1 I counted 90 subfamilies with 1,012,792 counts total. I am not sure to understand the connexion between the 52 clusters of 500 or more copies for a total of 1,157,133 copies. Thank you for clarifying. Another question about table S1 is about the origin of the p-value indicated? In addition,

this paragraph mentioned Table S2, but it correspond to the RNA-seq data. Finally, I did not see options with cd-hit-est to control the minimum sequence length in the clustering alignments, thus, is it possible that you have clusters made of small sequence overlaps such as:

```
seq 1 XXXXXXXXXXXXXXXXXXXMMMMMM
seq 2 MMMMMMMXXXXXXXXXXXXXXXXXX
```

For example, -aS can be used to force a minimum % of the shorter sequence to be included in the alignment. Note that cd-hit report the longest sequence as representative, not the consensus of the cluster.

3 - I really liked the intention of the authors to test for positive selection (local adaptation) on the young SINE elements. However, a simple Fst cutoff of the top 1% seems to me very hazardous. I understand that the goal is to generate a list of potential candidates, however, given the very strong population structure, in part due to artificial selections, it is expected to observe high level of variation in the measure of the pairwise differentiation at individual loci. At minimum, the outliers should be compared to an estimation of the "neutral expectation" generated from the distribution of the observed data and the population structure. These reference might help the authors choosing a well-suited test for their model: <https://academic.oup.com/zoolinnean/article/184/2/528/4970495>, <https://www.nature.com/articles/s41598-020-76576-3>, <https://onlinelibrary.wiley.com/doi/full/10.1111/mec.14184>.

Minor/Typos:

- 1- l.89 "bird" is too vague
- 2- l.102 silence → silencing
- 3- l.144 - TE → TE copies (insert in the genome)
- 4- Fig 5. The female symbol is inclined 45 degree to the right; I also discourage using stereotypic blue (male) and pink (female) colours.
- 5- l.297: Staufen-mediated decay is a possible explanation, but more generally a SINE or other insert in 3'UTR may affect transcript stability with other means (for example simply by increasing the length of the UTR: <https://doi.org/10.1098/rstb.2019.0331>).
- 6- l.313: enrichment → enriched
- 7- Can the author provide the exact version of RepeatModeler used?
- 8- KEGG/GO analysis: why using all human genes as background, and not the 14,000+ PCGs from the studies?
- 9- Which version of MELT was used?

Reviewer #2 (Remarks to the Author):

In this manuscript, Zhao et al. aim at assessing the diversity of transposable elements present in the pig genome and, for the SINEs that are polymorphic across a large sample of pig genomes, their functional consequences. To this end, the authors perform a new annotation of TEs in the pig genome through a combination of similarity-, structure-, and de novo-based methods and find that SINEs contribute the vast majority of relatively recent TE sequences (<20Mya). By cross-analysis of chromatin signatures (histone modifications, DNA methylation, accessibility, TF binding sites) together with tissue-specific transcriptomic datasets, they establish that the more recent SINEs are mostly depleted from active chromatin regions (as is expected given the strong purifying selection that acts on TE polymorphisms). For the few young SINEs that remain in genic regions and that are associated

with gene transcripts, they appear to contribute to gene expression by affecting the UTRs of genes with high tissue-specificity. Furthermore, they identify >200k SINEs that are polymorphic across >300 pigs covering the majority of Eurasian breeds and detect a few that show strong population-specific differentiation suggestive of a potential contribution to local adaptation. Finally, they identify 54 candidate genes where a SINE polymorphism is in linkage with a SNP identified by GWAS to be associated with a complex trait.

The state-of-the-art analyses and results presented in this manuscript by Zhao et al. establish a very rich and clear atlas of TEs in the pig genome and the potential of SINEs to contribute to gene expression and local adaptation. The quality of the datasets assembled here is illustrated in particular by the very clear clustering between genetic groups based on polymorphic SINE alleles and as a result the authors are able to identify a number of interesting candidate genes where a SINE insertion could be contributing a functionally relevant trait. However, the study falls short of demonstrating this functional relevance, even for one or two of their more promising candidates, which would greatly boost the relevance of the manuscript. For example, the polymorphic SINE identified in the first exon of RUNX3 appears to be polymorphic among Luchuan pigs, which could be used to confirm phenotypically its impact. Furthermore, the authors focus on polymorphic SINEs that are in linkage with trait-associated SNPs and therefore miss the potential of these polymorphic SINEs to be themselves associated with traits without being tagged by SNPs due to their young age (as was found to be the case for GWAS of polymorphic TEs in tomato for example). As a result, this study provides an in-depth catalog of TEs with a number of interesting candidates which will be of great interest to the community but that remains so far purely associative. Therefore, the authors must tone down some of their conclusions before publication, most notably in the title where the role of TEs in complex traits is not demonstrated.

Minor comments:

l.72 "or" : typo

Fig. 6: the colors do not match between 6b and 6e which makes the figure hard to follow

Discussion: numerous typos e.g. l. 514 "were more significant enrichment" or l. 548 "that are potential to contribute"

Reviewer #3 (Remarks to the Author):

Pigs are an important agricultural species with an interesting history of domestication. A large wealth of genomics resources exist, yet analysis of this data has been incomplete. Here, the authors perform a large scale assessment of transposable element dynamics in the pig genome, with a focus on a specific group of SINEs. The amount of genomic data interrogated is impressive and there are several potentially intriguing findings. However, the main message of the manuscript is hard to discern given the number of analyses described without a clear linking narrative or rationale. The manuscript would benefit from a more succinct description of confirmatory or expected findings and by providing additional detail related to novel claims.

Overall, novel terminology is unnecessarily introduced. For example, the term "polySINE" seems to be equivalent to what is generally known as a dimorphic SINE in the mobile element literature. The category names of "young" "younger" and "youngest" "older" and "oldest" as family names are confusing and could be replaced by clearer designations for each lineage. The abbreviation PCG (protein coding gene?) is never defined and does not seem necessary. The analysis of SINE derived transcripts is a core part of the paper and this analysis requires clarification.

Comments on sections of the manuscript:

Composition of young SINE families in the pig genome. Much of this description is consistent with that found in other annotated mammals. It could be greatly condensed to focus on the apparently young group of SINEs, since this is the focus of the remainder of the manuscript.

Analysis of ATAC-seq data (lines~201). Some details should be described as to how the analysis was done. Is it possible to uniquely map ATAC-seq reads to young SINEs? How is this handled in the analysis? Could the association with age simply reflect that there are more differences among older SINEs and thus it is easier to map the reads? Additional detail on the chromatin marks should be provided as well. I take it this is based on SINE presence in called peak regions rather than in a mapping of chipseq reads to element consensus sequences?

Line 242 The description of transcription factor binding sites present in pig SINE families is interesting but the description could be more clear. How strong is the match to the reported binding site? How conserved is this region among the SINE families/sequences? The connection to ZNF148 is intriguing but is not elaborated on. In humans, ref 72 shows that this gene is involved in muscle development regulation. How similar is the pig ZNF148 gene in terms of sequence, promoter, expression pattern, etc. How common is the predicted binding site? What genes are near the predicted binding sites? Etc. This observation required further elaboration. Similarly, the RFX1/RFX3 examples require further elaboration.

Line 251: Do you mean that SINEs themselves are depleted for CpG sites, or that they are located in genomic regions that are depleted from CpGs? Similarly, how confident is the mapping of methylation data to individual SINEs given the many copies that are present in the genome?

SINE-derived transcriptomes. This section is potentially interesting but is hard to follow. The terms are not clearly defined (i.e, what is exon covered?). Figure S9 does not have labels clearly explained— what are the red boxes? What does it mean that transcribed are inserted by full length SINEs (line 277)? Does this mean that the ISO-SEQ data detects many transcripts that are just from SINEs? This could be more clearly distinguished from other transcripts that happen to include SINE sequences as described in part below.

Line 290. What is being associated? Is this a comparison of the expression levels of annotated genes and of RNA from SINEs (which ones)? The following text on SINE derived exons suggests that these are also the same genes. Overall, I had trouble understanding the message of this paragraph.

Line 298. The discussion of placement of SINEs in 5'/3' UTRs, internal exons, etc seems better placed before the above discussion of expression correlation. A succinct table that summarizes the number and type of SINE derived transcripts (such as location of SINE in the transcript) might help clarify the description.

Line 310. Not sure that the new term "young-D SINE" is required. And what is meant by SINE-derived transcripts is unclear.

The message behind the section starting at 317 is also unclear. I take it that SINE derived transcripts (however they are defined) show tissue specific patterns. Again, the description here is hard to follow. Is the idea that there are specific SINEs that are expressed only in different tissues? Or are these just transcripts that happen to contain SINE sequence in them? Wouldn't you expect some spurious expression in the neighborhood of expressed genes due to the open chromatin? The paragraph in line 330 makes a strong claim for selection and domestication but I again have trouble connecting the dots

since it is not clear what SINE-derived transcripts are.

The section on dimorphic SINEs shows that the genotypes recapitulate known population admixture and ancestry groupings, which makes sense and indicates that the SINE genotyping is performing well. However, the connection to adaptation is rather indirect (line 382) – as far as I can tell these are just dimorphic SINEs that happen to have high frequency differences (as many thousands of SNPs must) or are near gene or regions otherwise associated with a trait of interest. However, there seems to be little or no evidence that the SINEs themselves are the driving or causal factor in the associations. The same concern applies to the analysis of dimorphic SINEs in regions of complex trait associations. The claims of local adaptation and links to GWAS signals are speculative. It is not clear that there is any data indicating that the SINEs themselves are causal, rather than being another marker that is associated in the region.

Other specific comments

Line 90: "wide-ranged" should be "wide ranging"

Line 102: "the silence" should be "the silencing"

Line 104-106: What is the significance of referencing angiosperms in the introduction?

Line 105: why are plants relevant to the introduction to this paper? Methylation is quite distinct in plants.

Line 150-151. The wording here is unclear. What does "65 consisted of 84.6% classified TEs" mean? Does that mean that 65 of the 532 families are large and account for 84% of all annotated elements?

Line 288: "felled" Does this mean falls or located in an intron?

Line 537-538: Should there be a reference for this?

Figure 7a: Fst label on the x-axis is blocked by figure panel below

Figure 8c: Breeds is labelled for 8d/g but not a or c.

Reviewers' comments:

Reviewer: 1

Comments to the Author

I have now reviewed the manuscript of Zhao et al. entitled "Building an atlas of transposable elements reveals the extensive roles of young SINE in gene regulation, genetic diversity, and complex traits in pigs". In their article, the authors perform an in-depth analysis of the transposable elements (TEs) distribution and potential functions across a vast number of pigs using a impressive amount of complementary data including 14 genome assemblies, hundreds of WG-resequencing, transcriptomics and epigenomics dataset.

After a thorough description of the overall TE content, the authors focus on the most abundant SINEs elements, which also includes the most evolutionary recent TE families in pigs. Younger SINEs seem associated with increased silencing (using different measures of chromatin accessibility, nucleosome location, and histone marks) while some rare instances seem to escape repression and appear to contribute in diverse fashions to the gene regulatory network. Further analyses connect these TEs with functional polymorphisms indicated by their co-segregation with variants associated with complex traits. Population genetics analyses using the segregating SINES (poly-SINES) recapitulates known population structure, and some SINEs are also suspected to show signature of local adaptation.

I found the manuscript well written and the succession of experiments easy to follow. As for the relevance of the experiments, I believe that some aspects of several analyses (as described in the supplementary data) need to be addressed. My general comment is that though general trends can be legitimately drawn from the sum of all the analyses, some individual results may require improved statistical evaluation, or further justification from the authors.

AU: We are incredibly grateful for the time you took to review our manuscript. Your insightful comments and suggestions have been invaluable in helping us improve our work. We have taken great care to address each of the issues you raised point-by-point, and have made revisions to our manuscript accordingly.

Main:

1 - "Definition of chromatin status and calculation of fold enrichment". An important part of these results rely on the differential enrichment of the SINEs in different functional compartments of the genome according to their relative age group. While the authors present the fold-enrichment for each functional region in each of the 14 considered tissue, each enrichment does not appear to be formally tested. In other words, do the observed enrichments calculated in each tissue deviate significantly from the expectation if TE and functional regions were distributed randomly? This could be addressed by shuffling TE positions across the genome. You can for example use this

script: https://github.com/4ureliek/TEanalysis/blob/master/TE-analysis_Shuffle_bed.pl

(you can access the full description by calling the help or reading in the code directly. Note that I am not the author of this script).

AU: Thank you very much for your suggestions. We have previously used `regionR` (`regionR`: Association analysis of genomic regions based on permutation tests) to test for enrichment in functional regions with the permutation test, and the results were consistent with the fold-enrichment reported in our manuscript. However, as per your suggestions, we have now formally tested these results and have included them in the revised manuscript on Line 199-202, as well as in Figure 2d and the Methods on Line 656-657.

Figure 1. Z-score and fold-enrichment for the enrichment of TE in functional regions

2 - "Pre-processing of young SINEs"

2.1 - It was unclear how nested SINE copies were dealt with. Starting line 70 of the supplementary data, "`parseRM_GetNesting.pl`" was used to remove the nesting and nested young SINEs, followed by a length filtering to remove elements with length > 200 bp and < 300 bp using a shell script.": did instances of SINE with an embedded TE, or a SINE within another element were removed? Why not using the nested SINEs?

AU: Thank you for pointing out our mistake. We apologize for our carelessness in writing the

methods section. The length of SINE actually ranges from 200 to 300 bp, and we meant to say that transposable elements within this length range are preserved. We have corrected this statement in the revised manuscript on supplementary methods. Regarding the removal of nested TEs, the purpose is to prevent other types of TEs (such as LINE) from affecting the enrichment of downstream functional regions (such as histone modifications).

2.2 line 73: "52 of 17,294 SINE clusters (n > 500) involving 1,157,133 young SINEs" -- In table S1 I counted 90 subfamilies with 1,012,792 counts total. I am not sure to understand the connexion between the 52 clusters of 500 or more copies for a total of 1,157,133 copies. Thank you for clarifying. Another question about table S1 is about the origin of the p-value indicated? In addition, this paragraph mentioned Table S2, but it corresponds to the RNA-seq data.

AU: Thank you for bringing our mistake to our attention. We apologize for placing Tables S1 and S2 in the wrong sections of our Supplementary Information. This led to confusion between the "52 clusters" and "90 subfamilies". Please note that Table S1 represents the subfamily classification by the COSEG pipeline, not the simple clustering by CD-HIT-EST. The output of the COSEG pipeline includes some SINEs with no subfamily classification, so only 1,012,792 counts are less than 1,157,133. This is why Table S2 corresponds to the RNA-seq data. We have corrected these errors in our revised Supplementary Information.

Finally, I did not see options with cd-hit-est to control the minimum sequence length in the clustering alignments, thus, is it possible that you have clusters made of small sequence overlaps such as:

```
seq 1 XXXXXXXXXXXXXXXXXXXMMMMMMM
```

```
seq 2 MMMMMMMXXXXXXXXXXXXXXXXXX
```

For example, -aS can be used to force a minimum % of the shorter sequence to be included in the alignment. Note that cd-hit report the longest sequence as representative, not the consensus of the cluster.

AU: Thank you for your valuable suggestion. It is true that the situations you listed are possible. However, it is important to note that we take great care in completing the final TE family clusters through a rigorous two-step process. Firstly, we use cd-hit-est to perform a preliminary clustering, which helps to narrow down the potential clusters. Then, in the next step of the process, the COSEG pipeline is employed to filter out any clusters that do not meet our strict criteria for inclusion. This ensures that the final TE family clusters are both accurate and reliable, providing a solid foundation for our ongoing research efforts.

3 - I really liked the intention of the authors to test for positive selection (local adaptation) on the young SINE elements. However, a simple Fst cutoff of the top 1% seems to me very hazardous. I understand that the goal is to generate a list of potential candidates, however, given the very strong population structure, in part due to artificial selections, it is expected to observe high level of variation in the measure of the pairwise differentiation at individual loci. At minimum, the outliers

should be compared to an estimation of the "neutral expectation" generated from the distribution of the observed data and the population structure. These references might help the authors choosing a well-suited test for their model:

<https://academic.oup.com/zoolinnean/article/184/2/528/4970495>

<https://www.nature.com/articles/s41598-020-76576-3>

<https://onlinelibrary.wiley.com/doi/full/10.1111/mec.14184>

AU: Thank you for your guidance. We neglected the "neutral expectation" and will add it to our revised manuscript. We have selected the BayeScan program (v2.1) and cited models to support our original results, as you recommended. The BayeScan program can be used to test for the deviation of each locus from neutrality. However, the signals are calculated between Cluster I and the remaining clusters. Since the remaining clusters may include multiple different populations and have a huge size difference, the BayeScan test may yield weak signals. To address this, we combined the previous F_{st} values with the BayeScan test to provide a reference for potential candidates. We have corrected this section in our revised manuscript (Line 332-334), TableS6, and the methods (Line 641).

Minor/Typos:

AU: Thanks for pointing out our mistake. We have checked all the minor comments and corrected them in the revised manuscript.

1- 1.89 "bird" is too vague

AU: We have removed it from the revised manuscript.

2- 1.102 silence → silencing

AU: We have corrected "silence" to "silencing" in the revised manuscript (Line 94)

3- 1.144 - TE → TE copies (insert in the genome)

AU: We have corrected "TE" to "TE copies (insert in the genome)" in the revised manuscript (Line 135)

4- Fig 4. The female symbol is inclined 45 degree to the right; I also discourage using stereotypic blue (male) and pink (female) colours.

AU: Thanks for your kindly remark. We have corrected it in the Figure 4 according to your requirements.

5- 1.297: Staufen-mediated decay is a possible explanation, but more generally a SINE or other insert in 3'UTR may affect transcript stability with other means (for example simply by increasing the length of the UTR: <https://doi.org/10.1098/rstb.2019.0331>).

AU: Thanks for your insightful suggestion. We have added this new possible explanation and cite in our revised manuscript (Line 262-264).

6- 1.313: enrichment → enriched

AU: We have corrected “significant enrichment” to “significantly enriched” in the revised manuscript (Line 291).

7- Can the author provide the exact version of RepeatModeler used?

AU: Thank you for your kindly remarks. We have added the exact version for RepeatModeler (open-1.0.11) in the revised Supplementary methods.

8- KEGG/GO analysis: why using all human genes as background, and not the 14,000+ PCGs from the studies?

AU: Thank you for your question. Regarding the functional enrichment analysis, we transformed pig genes into human genes. This was done because human KEGG annotations are more complete than those of pigs, and it allows for obtaining more SINE-related functional pathways. In our study, the functional enrichment analysis of genes is mainly aimed at exploring genes that are co-expressed with SINE-related genes. Therefore, only 2,744 genes were used for this enrichment analysis.

9- Which version of MELT was used?

AU: Thank you for your kindly remarks. We have added the exact version for MELT (version 2.2.2) in the revised manuscript (Line 462-463).

Reviewer: 2

Comments to the Author

In this manuscript, Zhao et al. aim at assessing the diversity of transposable elements present in the pig genome and, for the SINEs that are polymorphic across a large sample of pig genomes, their functional consequences. To this end, the authors perform a new annotation of TEs in the pig genome through a combination of similarity-, structure-, and de novo-based methods and find that SINEs contribute the vast majority of relatively recent TE sequences (<20Mya). By cross-analysis of chromatin signatures (histone modifications, DNA methylation, accessibility, TF binding sites) together with tissue-specific transcriptomic datasets, they establish that the more recent SINEs are mostly depleted from active chromatin regions (as is expected given the strong purifying selection that acts on TE polymorphisms). For the few young SINEs that remain in genic regions and that are associated with gene transcripts, they appear to contribute to gene expression by affecting the UTRs of genes with high tissue-specificity. Furthermore, they identify >200k SINEs that are polymorphic across >300 pigs covering the majority of Eurasian breeds and detect a few that show strong population-specific differentiation suggestive of a potential contribution to local adaptation. Finally, they identify 54 candidate genes where a SINE polymorphism is in linkage with a SNP identified by GWAS to be associated with a complex trait.

The state-of-the-art analyses and results presented in this manuscript by Zhao et al. establish a very rich and clear atlas of TEs in the pig genome and the potential of SINEs to contribute to gene expression and local adaptation. The quality of the datasets assembled here is illustrated in particular by the very clear clustering between genetic groups based on polymorphic SINE alleles and as a result the authors are able to identify a number of interesting candidate genes where a SINE insertion could be contributing a functionally relevant trait.

AU: Thank you for reviewing our manuscript. We appreciate your valuable comments and suggestions, which will help improve our articles.

However, the study falls short of demonstrating this functional relevance, even for one or two of their more promising candidates, which would greatly boost the relevance of the manuscript. For example, the polymorphic SINE identified in the first exon of RUNX3 appears to be polymorphic among Luchuan pigs, which could be used to confirm phenotypically its impact. Furthermore, the authors focus on polymorphic SINEs that are in linkage with trait-associated SNPs and therefore miss the potential of these polymorphic SINEs to be themselves associated with traits without being tagged by SNPs due to their young age (as was found to be the case for GWAS of polymorphic TEs in tomato for example). As a result, this study provides an in-depth catalog of TEs with a number of interesting candidates which will be of great interest to the community but that remains so far purely associative. Therefore, the authors must tone down some of their conclusions before publication, most notably in the title where the role of TEs in complex traits is not demonstrated.

AU: Thank you for your valuable feedback. We have carefully reviewed all the issues you raised and made revisions to our manuscript accordingly. As per your suggestion, we have toned down some of the conclusions in our revised manuscript. In this study, we aimed to highlight the important role of TE in pigs. However, in future studies, we plan to investigate specific TEs/SINEs in greater depth to elucidate their functional significance.

Minor comments:

l.72 “or” : typo

AU: Thanks for pointing out our mistake. We have corrected it in the revised manuscript.

Fig. 6: the colors do not match between 6b and 6e which makes the figure hard to follow.

AU: Thanks for your kindly remark. We have changed the color of the figure 6b to make it easier to understand.

Discussion: numerous typos e.g. l. 514 “were more significant enrichment” or l. 548 “that are potential to contribute”

AU: Thanks for pointing out our mistake. We have corrected all of them in the revised manuscript (Line 177).

Reviewer: 3

Comments to the Author

Pigs are an important agricultural species with an interesting history of domestication. A large wealth of genomics resources exists, yet analysis of this data has been incomplete. Here, the authors perform a large-scale assessment of transposable element dynamics in the pig genome, with a focus on a specific group of SINEs. The amount of genomic data interrogated is impressive and there are several potentially intriguing findings. However, the main message of the manuscript is hard to discern given the number of analyses described without a clear linking narrative or rationale. The manuscript would benefit from a more succinct description of confirmatory or expected findings and by providing additional detail related to novel claims.

AU: Thank you for reviewing our manuscript. We appreciate your valuable comments and suggestions. We have carefully addressed all the issues raised point by point and revised our manuscript accordingly. As per your suggestion, we have provided a more concise description of confirmed or expected results in our revised manuscript.

Overall, novel terminology is unnecessarily introduced. For example, the term “polySINE” seems to be equivalent to what is generally known as a dimorphic SINE in the mobile element literature. The category names of “young” “younger” and “youngest” “older” and “oldest” as family names are confusing and could be replaced by clearer designations for each lineage. The abbreviation PCG (protein coding gene?) is never defined and does not seem necessary. The analysis of SINE derived transcripts is a core part of the paper and this analysis requires clarification.

AU: Thank you for your constructive suggestions. As per your suggestion, we have replaced all instances of "polySINE" with "dimorphic SINE." We have also removed some abbreviations, such as "PCG," and replaced them with their full names, such as "genes." However, since the young SINE family consists of multiple families, we have kept the term "young SINE" for ease of illustration.

Comments on sections of the manuscript:

Composition of young SINE families in the pig genome. Much of this description is consistent with that found in other annotated mammals. It could be greatly condensed to focus on the apparently young group of SINEs, since this is the focus of the remainder of the manuscript.

AU: Thanks a lot for your guidance. We have shortened some descriptions about the composition of SINE families (Line 128-171).

Analysis of ATAC-seq data (lines~201). Some details should be described as to how the analysis was done. Is it possible to uniquely map ATAC-seq reads to young SINEs? How is this handled in the analysis? Could the association with age simply reflect that there are more differences among older SINEs and thus it is easier to map the reads? Additional detail on the chromatin marks should

be provided as well. I take it this is based on SINE presence in called peak regions rather than in a mapping of chipseq reads to element consensus sequences?

AU: Thank you for your kind remarks. The results and methods of the ATAC-seq analysis were cited from the FR-AgENCODE project (<http://www.frangencode.org/results.html>). Additionally, details of the chromatin mark analyses have been added to the methods (Line 541-549). Regarding the question about older SINEs being easier to map, this may be a limitation due to short reads that cannot be eliminated. However, the mapped reads by Bowtie 2 were further filtered by $MAPQ \geq 10$ to obtain as many uniquely aligned reads as possible.

Line 242 The description of transcription factor binding sites present in pig SINE families is interesting but the description could be more clear.

How strong is the match to the reported binding site? How conserved is this region among the SINE families/sequences?

The connection to ZNF148 is intriguing but is not elaborated on. In humans, ref 72 shows that this gene is involved in muscle development regulation. How similar is the pig ZNF148 gene in terms of sequence, promoter, expression pattern, etc. How common is the predicted binding site? What genes are near the predicted binding sites? Etc. This observation required further elaboration.

Similarly, the RFX1/RFX3 examples require further elaboration.

AU: We are so sorry for our unclear statements. We have changed the description for this paragraph (Line 212-222).

- (1) We set the parameters of fimo software as “p-value=1e-4”, “--max-stored-scores 100000000” and “--alpha=1”. We have added the parameters to the methods. (Line 563-564). In our study, TFBS were predicted to have the binding motifs in at least one SINE family. The shared TFBSs were scanned by fimo software with the same parameters are highly conservative.
- (2) As your suggested. We have expanded the description for the ZNF148 gene between the human and pig in our revised manuscript (Line 217-218).
- (3) The description of the RFX1/RFX3 example has also been expanded in our revised manuscript (Line 220-222).

Line 251: Do you mean that SINEs themselves are depleted for CpG sites, or that they are located in genomic regions that are depleted from CpGs? Similarly, how confident is the mapping of methylation data to individual SINEs given the many copies that are present in the genome?

AU: Thanks for pointing out our unclear statements. What we want to say is that SINEs are in genomic regions that are depleted from CpGs. We have corrected this description for it in the revised manuscript (Line 226-227).

The question of confidence in mapping methylation data is insightful. In this study, we first mapped reads to the genome to identify whole genome-wide methylation levels. We then compared the enrichment of different TE families in regions with high methylation levels. Mapping methylation data to repetitive regions, like mapping Chip-seq reads, is a difficult problem that cannot be ignored. Long-read sequencing technologies, such as ONT long reads, may be a good choice for this purpose. In our study, to better assess the relationship between TEs and methylation, we used Bismark with the "--ambig-bam" option to map the reads uniquely to the genome. This option only reports the first read with the best score, and if a read maps to multiple locations with the same score, it will be randomly aligned. As you reminded us, we have added the Bismark parameters to the methods section (lines 572-573).

SINE-derived transcriptomes. This section is potentially interesting but is hard to follow. The terms are not clearly defined (i.e, what is exon covered?).

AU: We are so sorry for our unclear statements.

We have completely revised the statement of this paragraph, including the description of SINE-derived transcripts and the structure of the paragraph. This paragraph consists of two parts:

The first part is the annotation of the SINE-derived transcripts and the description of the structural relationship between SINE and their derived transcripts (Line 246-259).

The second part included functional annotations of SINE-derived transcripts, mainly through enrichment analysis of their co-expressed gene (Line 261-298).

At last, we have defined some terms with clear descriptions as possible.

Figure S9 does not have labels clearly explained—what are the red boxes? What does it mean that transcribed are inserted by full length SINEs (line 277)? Does this mean that the ISO-SEQ data detects many transcripts that are just from SINEs? This could be more clearly distinguished from other transcripts that happen to include SINE sequences as described in part below.

AU: Thanks for pointing out our unclear statements.

- (1) The red boxes in figure S9 represented the SINEs and the yellow boxes represented the exons. We have completed the descriptions for some supplementary figures.
- (2) “The transcribed are inserted by full length SINEs” means some transcripts contain more than 80% of the sequence of the transcript. We have added the details to the revised methods (Line 596).
- (3) Sorry for our unclear definitions to “TE derived transcripts”. As you questioned, some transcripts totally derived from SINEs can be detected by ISO-SEQ data. But they have been removed in our TE derived transcripts. We have changed the descriptions for this part in the revised manuscript (Line 251-259).

Line 290. What is being associated? Is this a comparison of the expression levels of annotated genes and of RNA from SINEs (which ones)? The following text on SINE derived exons suggests that these are also the same genes. Overall, I had trouble understanding the message of this paragraph.

AU: Sorry for our mistake. There is some confusion here about association of annotated genes and RNA from SINEs. In fact, we intend to provide the comparison of the expression levels of annotated genes and of RNA from SINEs through co-expression. We have corrected this part in our revised manuscript (Line 278-280).

Line 298. The discussion of placement of SINEs in 5'/3' UTRs, internal exons, etc seems better placed before the above discussion of expression correlation. A succinct table that summarizes the number and type of SINE derived transcripts (such as location of SINE in the transcript) might help clarify the description.

AU: Thank you so much for your insightful guidance. As your suggested, we have replaced the discussion of SINEs in 5'/3' UTRs above discussion of expression correlation (Line 260-270). And we have added a succinct table for number and type of SINE derived transcripts to table S4.

Line 310. Not sure that the new term “young-D SINE” is required. And what is meant by SINE-derived transcripts is unclear. The message behind the section starting at 317 is also unclear. I take it that SINE derived transcripts (however they are defined) show tissue specific patterns. Again, the description here is hard to follow. Is the idea that there are specific SINEs that are expressed only in different tissues? Or are these just transcripts that happen to contain SINE sequence in them? Wouldn't you expect some spurious expression in the neighborhood of expressed genes due to the open chromatin? The paragraph in line 330 makes a strong claim for selection and domestication but I again have trouble connecting the dots since it is not clear what SINE-derived transcripts are.

AU: The term “young-D SINE” was removed as your suggested. And the detail description for the SINE-derived transcripts in the revised manuscript at Line 617-621.

The section on dimorphic SINEs shows that the genotypes recapitulate known population admixture and ancestry groupings, which makes sense and indicates that the SINE genotyping is performing well. However, the connection to adaptation is rather indirect (line 382) – as far as I can tell these are just dimorphic SINEs that happen to have high frequency differences (as many thousands of SNPs must) or are near gene or regions otherwise associated with a trait of interest. However, there seems to be little or no evidence that the SINEs themselves are the driving or causal factor in the associations. The same concern applies to the analysis of dimorphic SINEs in regions of complex trait associations. The claims of local adaptation and links to GWAS signals are speculative. It is not clear that there is any data indicating that the SINEs themselves are causal, rather than being another marker that is associated in the region.

AU: Special thanks for your insightful suggestions and comments. As you noted, it is currently

difficult to prove that dimorphic SINEs directly affect phenotypic traits. Our study aims to generate a list of potential candidates related to phenotypes, providing a research basis for follow-up investigations. In the future, we will attempt to perform functional verification for specific dimorphic SINEs and provide a complete TE association analysis, similar to GWAS.

To enhance the confidence of our results, we have changed the "sample fst cutoff of the top 1%" to a model called BayeScan. This model does a good job of considering the "neutral expectation" generated from the distribution of the observed data and the population structure.

Other specific comments

Line 90: "wide-ranged" should be "wide-ranging"

AU: Thank you for pointing out our mistake. We have corrected it in the revised manuscript. (Line 82).

Line 102: "the silence" should be "the silencing"

AU: Thanks a lot for your suggestion. We have corrected it in the revised manuscript. (Line 94).

Line 104-106: What is the significance of referencing angiosperms in the introduction?

Line 105: why are plants relevant to the introduction to this paper? Methylation is quite distinct in plants.

AU: Thank you very much for your recommendation. We have removed it in the revised manuscript.

Line 150-151. The wording here is unclear. What does "65 consisted of 84.6% classified TEs" mean? Does that mean that 65 of the 532 families are large and account for 84% of all annotated elements?

AU: Thanks for pointing out our unclear statements. We are so sorry for the confusion caused by our unclear statement. As your suggestion, we have changed this description at the revised manuscript (Line 140-141).

Line 288: "felled" Does this mean falls or located in an intron?

AU: Thanks for your kindly remark. We have corrected "located in" to "falls" in the revised manuscript (Line 288).

Line 537-538: Should there be a reference for this?

AU: Thanks for your suggestion. We have removed it in the revised manuscript.

Figure 7a: Fst label on the x-axis is blocked by figure panel below

AU: Thanks for your kindly remark. We have corrected it in Figure 7a.

Figure 8c: Breeds is labelled for 8d/g but not a or c.

AU: Thanks a lot for your suggestion. We have corrected it in Figure 8c.

Reviewers' comments:

Reviewer #1 (Remarks to the Author):

I have carefully reviewed the authors' response to the reviewers' comments. Most responses and subsequent changes greatly improved the quality of the manuscript. It remains a few points that need to be clarified before publication, that I have listed in two categories below:

Major:

- Thanks for complementing the statistical analysis of genomic and epigenomic feature enrichments using regionR. This is indeed a well-suited tool. Please provide the number of permutations in methods and p-values for the reported Z-scores.
- I appreciate the work done by the authors to improve their search for evidence of selection on dimorphic SINEs. Several aspects of the method used remains to be clarified. I found that Bayescan is indeed a relevant software to use however, i) the authors do not mention the use of Bayescan in the Results section. Moreover, the strategy used, in combination with the analysis of FST distribution is not clear. Please describe succinctly in Results and with more details in Method how candidates yielded from each method are combined. ii) "cluster i" isn't self-explanatory as introduced in the Results: please provide more information how the cluster are determined and what exactly is "cluster i". Next, it would be insightful to describe the rationale used to group all other clusters vs cluster i (e.g. significantly lower genetic variability between these groups or else). Finally iii), table S6 does not appear to be related to it as indicated in the response.
- Starting L.260: thanks for considering my suggestion regarding a possible regulatory role of SINEs in 3' UTR via non-specific sequence elongation (in contrast to SMD). However I think overall the section need to be reworked to improve clarity: I suggest the following plan: 1) the authors observed enrichment in young SINEs-derived transcripts in 3'UTRs. ii) Insertion into UTRs can affect transcript stability and gene expression, citing examples observed with data, iii) among possibles mechanistic explanations there are example of STD, which may be compatible in some instances of the observed data, and/or regulatory potential of increased UTR size. Choosing a mechanism would require more experiments, thus I suggest to remain cautious to avoid over-interpretation of the data.

Minor:

- 278: Would change "that clearly reflected tissue types" with "that is mostly consistent with tissue types"
- I would call REF- TE those absent from the ref, and REF+ TE present in the ref, i.e. the opposite of currently labelled.
- 407 "major" might be removed
- 416 add "[evolutionary] age"

Reviewer #3 (Remarks to the Author):

The revised manuscript incorporates many suggested changes from the reviewers and is improved. However, a sometimes-overwhelming amount of data is reported that makes the paper hard to follow and dilutes the main message. In particular, the analysis about chromatin patterns remains unclear.

It is still unclear how reduced read mapping may effect the chromatin enrichment statistics. In the

response document the authors states that a MAPQ threshold of 10 was used (I did not find this in the manuscript text itself). I think this could exasperate the issue – reads with a low MAPQ, such as those that align to young SINEs found in many copies in the genome – would be discarded. It must be clarified whether the signal shown in Figure 2 C, for example, simply reflects different mappability across SINE classes.

The description of transcripts that contain SINEs is improved, but I think still lacks clarity. If I understand, the analysis is of transcripts that happen to contain SINEs, with many of these being SINEs in UTR sequence. I think it is more accurate to call these SINE-containing transcripts rather than SINE-derived transcripts. I think SINE-derived implies that the transcription was initiated by the poI-III internal SINE promoter. Additionally, it is unclear whether the SINEs in analyzed transcripts are present in the reference genome or reflect population variation. How strong is the evidence that there are SINEs that contribute to the protein coding sequence of genes? Or are these rare insertions that would knock out the gene but they happen to be captured in the sample? Because of this I do not know how to interpret the statements in line ~269-270 about selection. If SINE was normally part of the gene/protein, wouldn't you expect that resulting transcripts that are sequenced to be "perfectly matched" with the annotated gene structure? Or is it that these are new SINE insertions that cause some sort of transcription problem resulting in only presence of single exon? Further clarification of these points and clear description of what is being analyzed would greatly aid the reader.

Relatedly, the sentence at line 250 that "This suggests that SINE plays an important role in regulating gene expression via TEs" does not make sense to me and the reasoning behind the claim is not clear.

The overall lack of clarity and difficulty in digesting the large amount of information presented is further reflected in the structure of the supplementary data. For example, Figures S1-S9 have meaningful legends, whereas Figures S10-S34 have only titles without explanation. The meaning of some aspects is not clear. For example, why are some bars red in Figures S20-S25? The title of Figure S30 appears to contain comments among the authors.

Typos/errors

Line 1 – title is not correct, should be "Genomes"

Line 135 '(insert in the genome)' I realize this is in response to a reviewer suggestion, but the literal copying of this text results in a sentence that does not make sense to me. Where would the TE copies be except inserted in the genome?

Line 150: Is active the correct word choice? Maybe most prolific or highest copy number. Although there are dimorphic elements I am not aware of evidence showing that the SINEs remain active (although I agree they probably are).

Line 176. Define TFBS on first usage

Line 228 "Similarly, We". W should not be uppercase.

Figure 4B. I am unclear on the data for this figure. How can something be an intron SINE but also UTR? Maybe I am misreading.

Figure S10 and others require explanation of what is being shown.

Line 291: Is M2 the same as ME2 (as shown in figure)?

Figure 6a: why is ref- the SINES that are in the reference and Ref+ are those that are not? I feel like this is unintuitive.

Line 316. The statements says that 85% of dimorphic sines are shared between groups. But I am unsure how Figure S28 shows that since only per-group frequencies are shown.

Figure 6 – Panel G not described in the legend

Figure S29 – without labels across the bottom to identify the samples it is not possible to interpret this figure.

Line 332 says bimorphic instead of dimorphic

Line 338-340 and 7b: The statements says that FRZB has a population specific dimorphic SINE, but Figure 7b shows histone mod data instead of anything about the population distribution of the insertion.

Figure 7b still refers to poly-SINEs

Line 343 Do the pigs with RUNX3 insertion have increased risk of T-cell malignancy or other diseases?

Line 349 What is "QTX"?

Figure 7D The figure would be clearer if the frequency plots had the same Y axis range.

Figure S30 Leftover bold text in legend appears to be comments from authors

Line 370. Why is this a hotspot? How are hotspots defined? Do the two mentioned genes have SINE insertions or are they just in the region?

Figure S32 Figure is hard to read. What are the coordinates shown?

Figure 8G does not seem to be mentioned in the body of the manuscript

Line 397. What does it mean for an exon to be "covered" by a SINE? Does this mean that the exon is 100% SINE sequence?

Figure 8i Should the SINE-derived orange line also be in the right side panel?

Line 411 "SINE is shorter and more complete than LINE". What does complete mean in this sentence? Does it refer to the common 5' truncation of LINES?

Line 411 "T" in "The" should not be upper case.

Line 414 Is L13 on subfamily or multiple subfamilies?

Line 520. Which version of minimap2 was used? There was a change in how large structural variants are aligned across introduced into new minimap2 versions so it is important to specify the version used.

Reviewers' comments:

Reviewer: 1

Comments to the Author

I have carefully reviewed the authors' response to the reviewers' comments. Most responses and subsequent changes greatly improved the quality of the manuscript. It remains a few points that need to be clarified before publication, that I have listed in two categories below:

AU: Thank you very much for taking the time to review our manuscript again. Your comments and suggestions have been extremely helpful in improving the quality of our work. We have carefully considered all of the points you raised and have made revisions throughout the manuscript to address them. We hope that these revisions demonstrate our commitment to producing high-quality research and that they address any concerns you may have had. Once again, we appreciate your time and expertise in reviewing our manuscript.

Major:

- Thanks for complementing the statistical analysis of genomic and epigenomic feature enrichments using regionR. This is indeed a well-suited tool. Please provide the number of permutations in methods and p-values for the reported Z-scores.

AU: Thank you for acknowledging our revision work. As per your requirements, we have included the number of permutations in the methods at line 661 in our revised manuscript. Additionally, we have added the Z-scores and their p-values in Table S2 of our revised supplementary tables.

- I appreciate the work done by the authors to improve their search for evidence of selection on dimorphic SINEs. Several aspects of the method used remains to be clarified. I found that Bayescan is indeed a relevant software to use however,

i) the authors do not mention the use of Bayscan in the Results section. Moreover, the strategy used, in combination with the analysis of FST distribution is not clear. Please describe succinctly in Results and with more details in Method how candidates yielded from each method are combined.
ii)"cluster i" isn't self-explanatory as introduced in the Results: please provide more information how the cluster are determined and what exactly is "cluster i". Next, it would be insightful to describe the rationale used to group all other clusters vs cluster i (e.g. significantly lower genetic variability between these groups or else). Finally iii), table S6 does not appear to be related to it as indicated in the response.

AU: i) Thank you for providing your valuable comments. Your insights will be important in our work. We strongly agree with your opinion on the importance of "neutral expectation" and the need to improve our understanding of transposon selection signals using the BayeScan program. However, the signals are calculated between Cluster I and the remaining clusters, which may include multiple different populations and have a significant size difference. Therefore, we found that the result of the BayeScan test only yielded weak signals in contrast to our original fst. The goal of the original fst was to generate a list of potential candidates for Cluster I with a specific allele frequency, and it achieved this goal. We hope to use the original fst model as the basis for subsequent analysis, and

we will supplement and display the results in the attached table (Table S8) with the results of the BayeScan program. In the future, we will primarily use information on "neutral expectation" to detect selection signals of polymorphic SINEs in larger populations. Thank you again for your suggestions.

ii) We apologize for the lack of clarity regarding cluster determination in the previous version of this document. In fact, we classified the different breeds into groups based on the optimal number of ancestral components, which was inferred to be $K = 10$ with the lowest CV (0.229). We have included this description in the revised manuscript (Line 328-331).

iii) Sorry for our mistake. We have added the BayeScan results to Table S8. We included the alpha values from BayeScan results in column eight.

- Starting L.260: thanks for considering my suggestion regarding a possible regulatory role of SINEs in 3' UTR via non-specific sequence elongation (in contrast to SMD). However I think overall the section need to be reworked to improve clarity: I suggest the following plan: i) the authors observed enrichment in young SINEs-derived transcripts in 3'UTRs. ii) Insertion into UTRs can affect transcript stability and gene expression, citing examples observed with data iii) among possible mechanistic explanations there are example of SMD, which may be compatible in some instances of the observed data, and/or regulatory potential of increased UTR size. Choosing a mechanism would require more experiments, thus I suggest to remain cautious to avoid over-interpretation of the data.

AU: We would like to express our gratitude for your valuable advice, which helped us realize our mistake. Your insightful comments made us aware that our previous description was unclear and may have caused confusion among our readers. Figure 4 illustrates the positional relationship between the SINE and its derived transcripts. To account for the impact of SINEs on derived transcripts that perfectly match annotated genes, we raise the possible effect of SINEs on gene expression with speculation and examples. Having considered your suggestion, we have revised the sentence once again, ensuring that there is no excessive interpretation of the data that may lead to any further confusion (Line 260-267). We appreciate your time and effort in providing us with such valuable suggestion.

Minor:

- 278: Would change "that clearly reflected tissue types" with "that is mostly consistent with tissue types"

AU: Thanks for pointing out our mistake. We have revised the manuscript to correct this sentence (Line 275).

- I would call REF- TE those absent from the ref, and REF+ TE present in the ref, i.e. the opposite of currently labelled.

AU: We apologize for any confusion regarding *Ref+* and *Ref-*. In our manuscript, we defined *Ref+* as the insertion of the TE to the reference and *Ref-* as the deletion of the TE from the reference. We have now added an explanation at line 1066-1067 in our revised manuscript to clarify the unclear

statements. Thank you again for your careful guidance.

- 407 "major" might be removed

AU: As you suggested, we have removed the word "major".

- 416 add "[evolutionary] age"

AU: As you suggested, we have added the word "evolutionary". (Line 417).

Reviewer #3 :

Comments to the Author

The revised manuscript incorporates many suggested changes from the reviewers and is improved. However, a sometimes-overwhelming amount of data is reported that makes the paper hard to follow and dilutes the main message. In particular, the analysis about chromatin patterns remains unclear.

AU: I'm glad that you approved of the work that I did. It gives me great pleasure to know that my efforts were worth it and that you are satisfied with the results. Your insightful comments and suggestions have been invaluable in helping us improve our work. We have extensively revised our manuscript, taking great care to incorporate your feedback in a meaningful way. We hope that you will find the revised version of our manuscript to be a significant improvement. Once again, we want to thank you for your time and expertise and for helping us make our work the best that it can be.

It is still unclear how reduced read mapping may affect the chromatin enrichment statistics. In the response document the authors states that a MAPQ threshold of 10 was used (I did not find this in the manuscript text itself). I think this could exasperate the issue – reads with a low MAPQ, such as those that align to young SINEs found in many copies in the genome – would be discarded. It must be clarified whether the signal shown in Figure 2C, for example, simply reflects different mappability across SINE classes.

AU: Thank you for your valuable comments. Your insights will be important for our future work. We share your concern that SINE mappability can vary depending on different SINE classes, and that this variation can have a significant impact that cannot be ignored. However, we believe that while mappability trends (e.g., older SINEs being easier to map the reads) are real, they are not the leading cause of the signal shown in Figure 2C. For instance, if only older SINEs with more sequence divergence are easier to map and lead to differences in chromatin signal, younger SINEs cannot exhibit high enrichment with MNase and H3K9me3 signals. Thanks to your reminder, we believe that in future studies we can explore the influence of chromatin enrichment of TEs in terms of mappability more specifically.

Regarding the statement "a MAPQ threshold of 10 was used," we apologize for our previous incorrect response. After carefully reviewing our previous procedures, we discovered that there was no MAPQ filtering step due to a communication error. In fact, our ChIP-seq analysis process uses the same batch of data (14 tissues) and results as described in our previously published article, "*Pig genome functional annotation enhances the biological interpretation of complex traits and human disease.*" The ChIP-seq data was aligned to the Duroc reference genome using BWA and peak calling was performed by MACS2. Thank you for your valuable comments.

The description of transcripts that contain SINEs is improved, but I think still lacks clarity. If I understand, the analysis is of transcripts that happen to contain SINEs, with many of these being SINEs in UTR sequence. I think it is more accurate to call these SINE-containing transcripts rather than SINE-derived transcripts. I think SINE-derived implies that the transcription was imitated by

the poI-III internal SINE promoter. Additionally, it is unclear whether the SINEs in analyzed transcripts are present in the reference genome or reflect population variation. How strong is the evidence that there are SINEs that contribute to the protein coding sequence of genes? Or are these rare insertions that would knock out the gene but they happen to be captured in the sample? Because of this I do not know how to interpret the statements in line ~269-270 about selection. If SINE was normally part of the gene/protein, wouldn't you expect that resulting transcripts that are sequenced to be "perfectly matched" with the annotated gene structure? Or is it that these are new SINE insertions that cause some sort of transcription problem resulting in only presence of single exon? Further clarification of these points and clear description of what is being analyzed would greatly aid the reader.

AU: Thank you for your valuable advice. Your questions helped us realize that our previous description was unclear and may have caused confusion for readers. This section mainly analyzes the transcripts that contain SINEs (SINE-derived), and Figure 4 illustrates the positional relationship between the SINE and its derived transcripts. Only to the transcripts with "perfectly matched" SINEs, we speculate and provide examples of the possible effect of SINEs on gene expression. We believe that the main ambiguity lies in this statement. We have revised the sentence once again, ensuring that there is no excessive interpretation of the data that may lead to any further confusion (Line 260-267). Regarding the selection discussed in the original manuscript on pages 269-270, we acknowledge that it provides a false comparison and overinterpretation. Therefore, we have removed this part of the statement. We appreciate your time and effort in providing us with such valuable suggestion.

Relatedly, the sentence at line 250 that "This suggests that SINE plays an important role in regulating gene expression via TEs" does not make sense to me and the reasoning behind the claim is not clear.

AU: Thank you for your suggestion. We agree that the claim was not clear and have revised this section in our manuscript. (Line 250-251)

The overall lack of clarity and difficulty in digesting the large amount of information presented is further reflected in the structure of the supplementary data. For example, Figures S1-S9 have meaningful legends, whereas Figures S10-S34 have only titles without explanation. The meaning of some aspects is not clear. For example, why are some bars red in Figures S20-S25? The title of Figure S30 appears to contain comments among the authors.

AU: Thank you for your guidance. Based on your suggestions, we have thoroughly revised the description and checked all the revised manuscript and supplementary files. We apologize for our carelessness and have corrected the error in Figure S30.

Typos/errors

Line 1 – title is not correct, should be "Genomes"

AU: Sorry for our mistake, we have corrected it in our revised manuscript. (Line 1)

Line 135 '(insert in the genome)' I realize this is in response to a reviewer suggestion, but the literal

copying of this text results in a sentence that does not make sense to me. Where would the TE copies be except inserted in the genome?

AU: Thank you for your suggestion. We completely agree with you and have deleted the sentence. Line 150: Is active the correct word choice? Maybe most prolific or highest copy number. Although there are dimorphic elements I am not aware of evidence showing that the SINEs remain active (although I agree they probably are).

AU: Thank you for your suggestion. We think "most prolific" is better, and we have changed it in line 152.

Line 176. Define TFBS on first usage

AU: Thank you for your suggestion. We define "TFBS" in lines 115-116.

Line 228 "Similarly, We". W should not be uppercase.

AU: Thank you for your suggestion. we have corrected it. (Line 228)

Figure 4B. I am unclear on the data for this figure. How can something be an intron SINE but also UTR? Maybe I am misreading.

AU: Sorry for the incorrect description. We have modified this section to improve clarity. (Line 260-267)

Figure S10 and others require explanation of what is being shown.

AU: Thank you for your suggestion. I have added a new description and explanation for Figure S10 and others in the revised supplementary information.

Line 291: Is M2 the same as ME2 (as shown in figure)?

AU: Sorry for our mistake, we have corrected it in our revised manuscript. (Line 288)

Figure 6a: why is ref- the SINES that are in the reference and Ref+ are those that are not? I feel like this is unintuitive.

AU: We apologize for any confusion regarding *Ref+* and *Ref-*. In our manuscript, we defined *Ref+* as the insertion of the TE to the reference and *Ref-* as the deletion of the TE from the reference. We have now added an explanation at line 1066-1067 in our revised manuscript to clarify the unclear statements.

Line 316. The statements says that 85% of dimorphic sines are shared between groups. But I am unsure how Figure S28 shows that since only per-group frequencies are shown.

AU: We apologize for incorrectly placing Figure S28 due to our modification of the article. We have changed it at line 313 in our revised manuscript.

Figure 6 – Panel G not described in the legend

AU: We apologize for our mistake and appreciate the reminder. We have changed legends for Figure 6g. (Line 1072-1073)

Figure S29 – without labels across the bottom to identify the samples it is not possible to interpret this figure.

AU: We apologize for our mistake. We have added the x-axis labels of figure S29 in our revised supplementary files.

Line 332 says bimorphic instead of dimorphic

AU: Sorry for our mistake, we have corrected it. (Line 331)

Line 338-340 and 7b: The statements says that FRZB has a population specific dimorphic SINE, but Figure 7b shows histone mod data instead of anything about the population distribution of the insertion.

AU: Sorry for the mistake of incorrectly removing a statement during our modification of the article. We have added it back in at line 338 of our revised manuscript.

Figure 7b still refers to poly-SINEs

AU: We are sorry for our carelessness, we corrected it in figure 7b.

Line 343 Do the pigs with RUNX3 insertion have increased risk of T-cell malignancy or other diseases?

AU: Thank you for your valuable question. Although there are no clear studies on the effect of RUNX3 on disease in the pig genome, it is certainly an issue that deserves further investigation. It is important to determine whether southern pig breeds and miniature pigs are associated with an increased susceptibility to certain diseases. It is crucial that we can continue to explore this topic in the future so that we may better understand the relationship between RUNX3 and disease.

Line 349 What is “QTX”?

AU: Sorry for our mistake, we have added full name for “QTX” in our revised manuscript. (Lines 348-349).

Figure 7D The figure would be clearer if the frequency plots had the same Y axis range.

AU: Thank you for your suggestion. These plots aim to highlight the frequency differences of specific SINEs within different populations. Since each SINE frequency is independent, we believe there is no need to adjust the Y-axis range of the frequency plots. Nevertheless, thank you again for your careful guidance.

Figure S30 Leftover bold text in legend appears to be comments from authors

AU: We apologize for my carelessness. We have removed the error and checked all the revised manuscript and supplementary files.

Line 370. Why is this a hotspot? How are hotspots defined? Do the two mentioned genes have SINE insertions or are they just in the region?

AU: We defined hotspots that have a higher rate of dimorphic SINEs than the surrounding region.

Figure S32 Figure is hard to read. What are the coordinates shown?

AU: We appreciate your careful assistance in improving our article. We have decided to remove figure S32 as it was difficult to understand and did not provide any useful information for interpreting our results.

Figure 8G does not seem to be mentioned in the body of the manuscript

AU: Thanks for your reminder. We have added it in our revised manuscript. (Line 396).

Line 397. What does it mean for an exon to be “covered” by a SINE? Does this mean that the exon is 100% SINE sequence?

AU: I'm really sorry for not being clear enough in my description. In fact, what we want to express is that the novel transcript covers the first exon of *VRTN* gene. We have corrected the description of this part in the revised version of the article. (Lines 396-398).

Figure 8i Should the SINE-derived orange line also be in the right side panel?

AU: Thank you for reminding us. We agree that the orange line should be added to Figure 8i. We have updated the right side panel of Figure 8i with an orange line.

Line 411 "SINE is shorter and more complete than LINE". What does complete mean in this sentence? Does it refer to the common 5' truncation of LINES?

AU: Thanks for your question. The word "complete" refers to SINEs that have all their sequences and are relatively intact, and are therefore fully integrated into the genome. This is in contrast to LINES, which can sometimes be truncated at the 5' end.

Line 411 "T" in "The" should not be upper case.

AU: Thanks for your reminder. We have corrected it in our revised manuscript. (Line 411).

Line 414 Is L13 on subfamily or multiple subfamilies?

AU: Thanks for your finding our mistake. L13 is single subfamily. We have corrected it in our revised manuscript. (Line 414).

Line 520. Which version of minimap2 was used? There was a change in how large structural variants are aligned across introduced into new minimap2 versions so it is important to specify the version used.

AU: According to your requirements, we have added the version for minimap2 in our revised manuscript. (Line 521).

REVIEWERS' COMMENTS:

Reviewer #1 (Remarks to the Author):

I am satisfied with the authors' response to my comments. Below is a list of small edits that I believe will improve clarity further.

309: Please add the description of Ref+ and Ref- as these terms are introduced in this section (in addition the addition already done in the figure legend).

329: the use of Bayscan need to be added here, especially if "alpha coefficient" is indeed from Bayescan. It should also be mentioned here what is the hypothesis: loci under selection should have a increased F_{st} (for the F_{st} approach) and alpha should be positive and significant (with Bayescan) for sign of positive selection. If the authors wish to relegate the Bayscan analysis in supplementary, it should be explained in the results section as well. I am in agreement with the explanation given in the response to my previous comments, and it can be used as rationale in the main text.

Figure 4A: the female symbol should points towards the bottom.

Figure 7A: I think it can be more impactful to show the whole distribution (F_{st} from 0 to 1) with the outliers colored in red for example. The gene label can be shown as a separate panel (zoom) or added in supplementary

Reviewer #3 (Remarks to the Author):

There are interesting and valuable aspects of this study, but it is disappointing that the authors appear to have not engaged seriously with the comments from the reviewers. The reviewers have spent substantial effort to understand and improve this study. Several aspects remain unclear. Here are three examples that demonstrate that the concerns of the reviewers have not been adequately addressed

1. Signals of selection

The previous review pointed out that claims of selection are confounded by neutral demographic processes and that without properly accounting for these processes there may be no evidence that any dimorphic SINEs show frequency patterns more extreme than would be expected by chance. BayeScan, a suggested a suggested analysis tool, was used that partially controls for this. Confusingly, the method is described in the methods section, but the results are not discussed. Apparently, this is because the "the result of the BayeScan test only yielded weak signals in contrast to our original". Omitting the result of statistical tests because they do not support the desired findings is not acceptable. This is not a valid reason to omit this analysis, and in fact may argue that there is no compelling evidence for selection.

2. Read mapping to repetitive sequence

The analysis of various read signals mapped to different repeat families is a major portion of the paper. Proper handling of reads that map to such sequences is critical for robust analysis. There are published examples of this issue leading to incorrect findings and there are studies describing ways to properly handle this issue (see PMID: 36451223, PMID: 31890048, PMID: 27046835, PMID: 25805138 among others). Since the initial submission it has been unclear how the authors performed these analyses. The latest response points to the methods of a previous paper (PMID: 34615879). The

methods of that paper state that "Briefly, the susScr11 genome assembly and Ensembl genome annotation (v100) were used as references for pig. Sequencing reads were trimmed with Trim Galore (v.0.6.5), and aligned with either STAR (v.2.5.4a) or BWA (v0.7.17) to the respective genome assemblies. Alignments with MAPQ scores <30 were filtered using Samtools (v.1.9)." Given the previous statements from the authors about mapping quality filtering it remains unclear how exactly the analysis was performed. The authors lack of familiarity with their own methods is a concern.

3. Terminology

This is a minor point, but it is illustrative of how the meaning and intent of previous reviews has not been incorporated. Despite comments in the initial review the unnecessary term "polySINE" is used again multiple times in the revised supplementary figures. Other confusing terminology issues remain.

Reviewer #4 (Remarks to the Author):

I was asked to focus my review on the comments raised by reviewer 3. Most points were adequately addressed in my opinion, but there are three issue which I think merit additional attention:

1. In response to comments related to read mappability, the authors made valid points and clarified how reads were mapped. I think it is important to state early on in the results section whether this mapping strategy is inclusive of multi-mapping reads or whether only unique reads were used.
2. I agree with the reviewer that 'SINE-derived transcript' is a misleading term that suggests transcript initiation from a SINE element, but this was not changed or counter-argued. I would suggest 'SINE-associated' or 'SINE-containing'.
3. I also agree with the reviewer that the nomenclature used for SINE transcripts can be difficult to follow, which led to the reviewer's comment that Figure 4B seemingly displayed a SINE on and intron and UTR at the same time. If I understand correctly, the x axis groups on this figure refer to how the transcript relates to annotated genes, whereas the colours refer to the position of the SINE within the transcript. If so, this need to be made clearer on the figure. Another suggestion would be to include a small schematic of how transcripts within each of these groups look like, to clarify what the nomenclature means.

Miguel Branco

Reviewers' comments:

Reviewer: 1

Comments to the Author

I am satisfied with the authors' response to my comments. Below is a list of small edits that I believe will improve clarity further.

AU: I am glad that you appreciate the work I have done. It is very satisfying to know that my efforts have been worthwhile and that you are pleased with the results. Your valuable insights and suggestions have been helpful in improving our work. We hope that you will find the revised version of our manuscript suitable for publication. We want to express our gratitude for your time, expertise, and for helping us to produce our best work.

309: Please add the description of Ref+ and Ref- as these terms are introduced in this section (in addition the addition already done in the figure legend).

AU: Thank you for your valuable advice. As your suggestion, we have added the new description for Ref+ and Ref- at line 305-307 in our revised manuscript.

329: the use of Bayscan need to be added here, especially if "alpha coefficient" is indeed from Bayescan. It should also be mentioned here what is the hypothesis: loci under selection should have a increased Fst (for the Fst approach) and alpha should be positive and significant (with Bayescan) for sign of positive selection. If the authors wish to relegate the Bayscan analysis in supplementary, it should be explained in the results section as well. I am in agreement with the explanation given in the response to my previous comments, and it can be used as rationale in the main text.

AU: Thank you for your guidance. Based on your suggestions, we have revised the description and added a hypothesis for the paragraph at line 325-330 in our revised manuscript. We have emphasized the alpha coefficient as a sign of positive selection in the results. We appreciate your time and effort in providing us with new software that helps us increase confidence in the detection of positive selection signals.

Figure 4A: the female symbol should point towards the bottom.

AU: Thanks for your reminder. We have corrected it in figure 4A of our revised manuscript.

Figure 7A: I think it can be more impactful to show the whole distribution (Fst from 0 to 1) with the outliers colored in red for example. The gene label can be shown as a separate panel (zoom) or added in supplementary

AU: Thank you for the reminder. We appreciate your suggestion to highlight high FST values as evidence of selection on the target genes. However, in Figure 7A, we intended to display FST values for multiple groups, and solely focusing on red outliers would not accurately reflect the classification of different groups. Therefore, we regret that we were unable to implement this suggestion. Nevertheless, we thank you for your valuable input, which has contributed to improving the overall quality of our article.

Reviewer: 4**Comments to the Author**

I was asked to focus my review on the comments raised by reviewer 3. Most points were adequately addressed in my opinion, but there are three issues which I think merit additional attention:

AU: Thank you for taking the time to review our manuscript. We appreciate your acknowledgment of our partial response. Your opinions and suggestions have been very helpful in improving the quality of our work. We have carefully considered all the issues you raised and made modifications throughout the manuscript to address them.

1. In response to comments related to read mappability, the authors made valid points and clarified how reads were mapped. I think it is important to state early on in the results section whether this mapping strategy is inclusive of multi-mapping reads or whether only unique reads were used.

AU: Thank you for your positive comments about our efforts. For the read mappability, this may be a limitation due to short reads that cannot be eliminated. What I am certain of is that our mapping strategy is to use only unique reads to minimize bias as much as possible. As you suggested, we have added the description at line 166 in our revised manuscript.

2. I agree with the reviewer that 'SINE-derived transcript' is a misleading term that suggests transcript initiation from a SINE element, but this was not changed or counter-argued. I would suggest 'SINE-associated' or 'SINE-containing'.

AU: Thank you so much for your insightful guidance. We apologize for misleading you with the question about the "SINE-derived transcript". As you suggested, we have replaced the SINE-derived transcript to SINE-associated transcript in our revised manuscript, Figures, Supplementary Information, and Supplementary Data.

3. I also agree with the reviewer that the nomenclature used for SINE transcripts can be difficult to follow, which led to the reviewer's comment that Figure 4B seemingly displayed a SINE on an intron and UTR at the same time. If I understand correctly, the x-axis groups on this figure refer to how the transcript relates to annotated genes, whereas the colours refer to the position of the SINE within the transcript. If so, this needs to be made clearer on the figure. Another suggestion would be to include a small schematic of how transcripts within each of these groups look like, to clarify what the nomenclature means.

AU: We apologize for our mistake in this part. To clarify, in Figure 4B, the x-axis groups represent how the transcript is related to annotated genes, while the colors indicate the position of the SINE within the transcript. As you suggested, we modified the figure and added a small schematic to make this clearer. Additionally, we added a description to the figure legend at lines 1214-1216 and lines 254-255 in our revised manuscript. We appreciate your time and effort in providing us with such valuable suggestions.